# OMNI-P2x universal neural network potential for excited-state simulations

Mikołaj Martyka [1], Xin-Yu Tong [2], Joanna Jankowska[1] ✉ & Pavlo O. Dral [2,3,4,5] ✉

Photo-active molecular systems play an essential role in modern science and technology, finding applications in solar cells, organic light-emitting diodes, reaction catalysis, photodynamic therapy, and beyond. The rational design of photo-responsive molecules requires an understanding of the photophysical and photochemical processes underlying their operation, which can be gained via the first-principles quantum-mechanical calculations that are prohibitively expensive for high-throughput investigations. To break through this limitation, here we introduce OMNI-P2x: a universal neural network potential for molecular excited and ground electronic states. OMNI-P2x can be used, directly or after fine-tuning, to perform a wide range of photophysical and photochemical simulations. OMNI-P2x approaches the accuracy of time-dependent density functional theory methods at a fraction of the computational cost, while being more accurate and faster than established semi-empirical methods for excited-state simulations. Here, we demonstrate its use in UV/Vis absorption spectroscopy, real-time photodynamical simulations, and in the rational design of visible-light-absorbing azobenzene systems.

Theoretical photochemistry leverages the ability of quantum-mechanical (QM) methods to accurately predict the properties of electronic excited states of molecules and to provide insights into light-molecule interactions. The QM calculations facilitate better understanding of complex processes such as photoreactivity and photostability[1–4], operation of molecular machines[5,6], separation of charge carriers in clean energy generation[7,8], and, in turn, provide rational design principles for photomaterials such as organic light-emitting diodes (OLEDs)[9,10], molecular solar thermal energy storage (MOST) systems[11,12], and organic solar cells[13,14]. Because of the extensive development of computational methods, they can be not only used to explain and verify experimental results, but also to guide novel research in photochemistry.

At the same time, the key obstacle preventing widespread use of theoretical methods for excited-state property prediction is their computational cost, with a schematic representation of their accuracy/computational cost tradeoff presented in Fig. 1. While highly-accurate methods, such as equation-of-motion coupled clusters approaches[15] and their variants like ADC(2)[16] exist, they can only be used for medium-sized systems at best, requiring days or weeks of computational time due to their steep computational scaling. On the other hand, time-dependent density functional theory (TD-DFT)[17,18] provides a more affordable alternative, which typically scales up to systems containing about 100 atoms, however, at a price of decreased reliability. Finally, on the lower end of the accuracy-to-computational-cost spectrum, one finds semi-empirical QM methods (SQM), such as a wide family of Hamiltonians based on the neglect of diatomic differential overlap (NDDO) approximation [19], such as AM1[20], PM3[21], OM2[22], and ODM2[23] used in conjunction with the configuration interaction (CI) treatments, and time-dependent tight-binding DFT methods TD-DFTB[24,25]. While these methods are certainly more affordable for systems with hundreds and even

[1]University of Warsaw, Faculty of Chemistry, Warsaw, Poland. [2]State Key Laboratory of Physical Chemistry of Solid Surfaces, College of Chemistry and Chemical Engineering, and Fujian Provincial Key Laboratory of Theoretical and Computational Chemistry, Xiamen University, Xiamen, China. [3]Institute of Physics, Faculty of Physics, Astronomy, and Informatics, Nicolaus Copernicus University in Toruń, Toruń, Poland. [4]Institute of Advanced Studies, Nicolaus Copernicus University in Toruń, Toruń, Poland. [5]Aitomistic, Shenzhen, China. ✉e-mail: jjankowska@chem.uw.edu.pl; dral@xmu.edu.cn

**Fig. 1 | The universal OMNI-P2x neural network potential for excited-state simulation and its place among the hierarchy of quantum mechanical (QM) and machine learning (ML) methods. a** OMNI-P2x and its fine-tuning shown in the context of computational cost/accuracy tradeoff of commonly used QM methods for excited-state simulations: semi-empirical quantum chemical methods (SQM), time-dependent density functional theory (TD-DFT), complete active space self-consistent field method and its derivatives (CASSCF), equations-of-motion coupled-clusters methods (EOM-CC), as well as active learning. **b** Overview of the OMNI-P2x design, training, and its application capabilities.

thousands of atoms, the accuracy of their predictions can often be questionable.[26–28]

The situation becomes even more complex when real-time simulations of the light-induced processes are required, i.e., for predicting photoswitching quantum yields, two-photon dynamics, or excited-state lifetimes. The most popular and widespread technique for performing such simulations is trajectory surface hopping (TSH)[29] variant of nonadiabatic molecular dynamics (NAMD). In TSH, a swarm of independent, semi-classical trajectories are propagated, with the ability to switch (hop) between adiabatic potential energy surfaces (PESs). The electronic and nuclear degrees of freedom are coupled by so-called interstate couplings, which control the hopping probabilities. Running surface-hopping simulations requires using a QM method that can accurately predict multiple PESs and, ideally, properly describe the multireference character of the wavefunction in the vicinity of conical intersections. This severely limits the choice of theoretical methods, and typically, CASSCF is used, while much slower but higher-accuracy perturbational variants such as CASPT2[30] are also available. However, as driving dynamics requires calculations of energies and gradients at each time step, with a typical simulation setup of hundreds of trajectories propagated for ~1 ps with a time step of 0.1–0.5 fs require computing these quantities on the order of millions of times, which is only possible for simple, often model systems. For simulations of experimentally and technologically relevant compounds, the only available choice are, so far, semi-empirical methods.

In recent years, progress in the field of machine learning (ML) has led to a significant paradigm shift in the area of computational chemistry.[31–35] Fitting machine learning potentials (MLPs) to reference QM data allows for accurate predictions of molecular properties, such as energies and energy gradients[36–40], dipole moments[39] or electronic densities[41,42] at a greatly reduced cost, while retaining the accuracy of the reference method. This allows to break through the limitation of traditional QM methods, making simulations of larger, more complex systems possible at longer timescales.[43–47] A particularly important, emerging class of potentials is universal MLPs, which we define as models that were trained on vast datasets spanning across chemical space. In turn, they can be used to make out-of-the-box predictions for previously unseen molecules, generalizing from the training data.[48–53]

Although training and using MLPs for predicting ground-state properties is becoming a rather routine task, using ML methods for electronic excited states is far more challenging, requiring expert human supervision. This is due to the inherent complexity of excited potential energy surfaces, often forming a strongly correlated manifold of states, interacting and mixing with each other in a non-smooth manner. At the same time, it should be noted that models capable of providing accurate predictions for excited-state properties, such as SchNarc[54], SpaiNN[55], or PyRAI²MD[44] have been developed and used successfully in the TSH simulations. However, their capabilities are limited to learning one molecule at a time, i.e., they are not transferable across chemical space.

This has changed with the MS-ANI (Multi-state ANI) model[56], demonstrated to be able to predict nonadiabatic dynamics of three distinct molecules within the same model, paving the way for universal excited-state potentials. More recently, another demonstration of a possibility to create transferable NN for excited-state predictions was shown in ref. 57. Transferable nonadiabatic molecular dynamics simulations were also reported by Axelrod et al.[58], however, it was limited to the class of diazobenzene photoswitches. Despite all the progress in MLPs for excited states, they have still been lagging behind the ground-state MLPs to such an extent that many experts are questioning whether universal MLPs for excited states are even possible.[59,60] It is generally agreed that developing such an MLP would be a major breakthrough.[31,59]

In this work, we present OMNI-P2x: a universal neural network (NN) potential for molecular excited states (Fig. 1). We have built OMNI-P2x by combining the ideas of multi-state[56,61] and all-in-one[62] learning. OMNI-P2x approaches TD-DFT accuracy for excited-state energies and targets CCSD(T) quality for ground-state energies: it can predict the absorption UV/Vis spectra out-of-the-box with higher speed and accuracy than popular semiempirical QM methods, making ML approaches for excited-states a viable alternative to established, cost-efficient QM methods for high-throughput calculations, such as semi-empirical methods.

We leverage the power of OMNI-P2x to perform a high-throughput screening of 500,000 compounds to find azobenzene derivatives featuring red-shifted, visible light absorption. These systems are particularly relevant due to their applications in nonlinear optics[63], photoresponsive polymers[64], and as MOST compounds[65,66]. The potential of OMNI-P2x in accelerating another type of extensive simulations: nonadiabatic dynamics, is also exploited via fine-tuning of this universal potential within an active learning loop. This approach reduces the computational cost and wall-clock time of production-ready ML-TSH simulations, while improving data efficiency and simulation stability.

## Results
### Model architecture and training
The architecture of the OMNI-P2x model is based on the idea of multi-state learning[56,61] and its successful realization in the MS-ANI model designed for accurate and efficient learning of excited-state energies and forces. Multi-state learning can be used to efficiently learn an arbitrary number of electronic states while using the entire neural network to differentiate the molecular electronic states, as opposed to only the last layer used in the multi-output models. This effect is achieved by including an additional state ordering number as an input feature, along with the molecular descriptor containing information

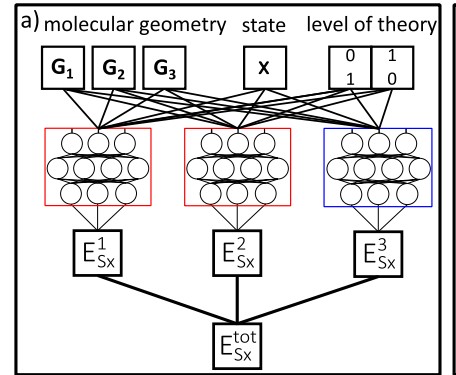
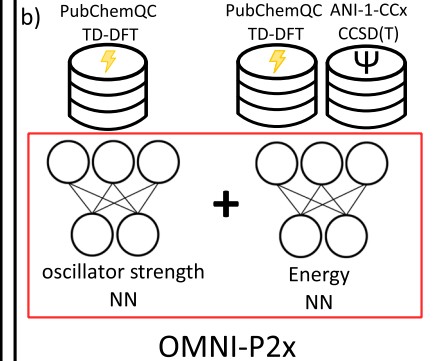

**Fig. 2 | Architecture of the OMNI-P2x model. a** The architecture of the OMNI-P2x neural network potential for energies. **b** Summary of the OMNI-P2x model, consisting of separate NNs predicting the oscillator strengths and energies of the target, and datasets used to train them, labeled at time-dependent density function theory (TD-DFT), or coupled-clusters with singles and doubles with perturbative triples (CCSD(T)).

about the geometrical conformation. Here, we extend this model to learn across multiple QM levels, in the spirit of the all-in-one (AIO) learning paradigm introduced recently[62]. The all-in-one learning has been successfully used to develop the universal OMNI-P1 neural network potential capable of making predictions approaching DFT and CCSD(T) levels, however, it is limited to the ground-state energies and forces. It is worth noting that the ability of learning from multiple levels of theory gives an important advantage of leveraging different data sets of various quantities. Here, we combine both: multi-state learning and all-in-one learning, to obtain OMNI-P2x: the omnipotential-2x, all-in-One Machine-learning Neural-network Interatomic Potential 2 for eXcited states.

OMNI-P2x is based on the established ANI NN potential[67] (Fig. 2a). The input layer consists of an atomic environment vector (AEV), calculated using the TorchANI Python package[68], and two additional features: the state ordering number, as in MS-ANI, and a one-hot encoded feature representing the QM level corresponding to all-in-one learning. The electronic state is encoded as a non-negative integer, consistent with the natural ordering of adiabatic state energies, which we find to be both physically motivated and empirically more accurate than one-hot encoding. In the present work, data at two levels of theory were included: the ground-state-only data with CCSD(T)/CBS accuracy from the ANI-1ccx dataset[67], and TD-DFT B3LYP/6-31+G* excited- and ground-state data from the PubChemQC dataset[69]. As for all ANI-type models, a separate neural network is trained for each chemical element. In the present version, OMNI-P2x supports the following element types: H, C, N, O, F, S, and Cl. The predictions are then made atom-wise and summed up to yield the total energy of the molecule. For the energy part of the predictions, an ensemble of three NN potentials is used, with their predictions averaged to obtain the final predicted energy for all electronic states. A fragment-correction scheme is applied to non-covalently bound structures, to recover the correct, intensive character of excitation energies, for molecular fragments separated by distances greater than the descriptor cutoff radius, with details provided in the methods section, and Suppl. Note. 1. A separate MS-ANI-type NN is trained and used to predict the oscillator strengths of interstate transitions, as shown in 2b). Since only TD-DFT data is used for the excited-state training, this network does not include an additional input feature for the level of theory.

The data for model training has been sourced from two datasets: 1) excited-state data in the PubChemQC project's dataset[69], and 2) ANI-1ccx, containing about 500,000 ground-state geometries, including out-of-equilibrium conformations, with their total energies computed at the gold-standard CCSD(T)/CBS level of theory. The PubChemQC project's dataset consists of unique, optimized structures of molecules at the TD-B3LYP/6-31+G* level of theory,[70,71] with the energies and

oscillator strengths of the first 10 excited states calculated using TD-DFT. We have used the entirety of the PubChemQC dataset, which meets our filtering criteria (as detailed in the Methods section), yielding 3.1 million molecules with 11 electronic states each. The inclusion of the ANI-1ccx data set provides an important additional information about the effect of out-of-equilibrium geometry changes on energies, which is important for molecular dynamics and other types of simulations. ANI-1ccx data set has been used as a basis for training multiple universal ML-based models[48,72–76], which are successfully used in molecular dynamics, geometry optimizations, and thermochemistry calculations. In addition, combination of two different types of data set allows to cover larger chemical spaces, i.e., compare the data distribution in both data sets in Fig. 3a: the ANI-1ccx data cover small-sized molecules, while the vast majority of the excited-state training data is comprised of medium-sized molecules containing 15 to 45 atoms, with most of their excitation energies located towards the blue end of the visible spectrum, and in the UV range (Fig. 3b).

We evaluate the performance of the model using a test set of 300k unseen molecules from the PubChemQC dataset for the excitation energies, as well as a test set of almost 49k ground-state conformations from the ANI dataset. The MAE in energy predictions for the first 11 electronic states with respect to the TD-DFT data shows that the quality of the predictions increases when going to higher excited states, reaching a minimal value for $S_7$ (Fig. 4b, c). An outlier here is $S_0$, which has the smallest energy MAE of 1.6 kcal/mol, which we attribute to the improved data-efficiency due to the application of the all-in-one learning to the combined set with the ground-state DFT (from Pub-Chem) and the CCSD(T) (from ANI-1ccx) data, and an overall smoother and lower-variance distribution of ground-state energies that makes them easier for the model to learn. This ground-state MAE can be compared to the similar metrics of other MLIPs: ANI-1[67] with an RMSE of 1.3 kcal/mol (OMNI-P2x 2.93 kcal/mol), UMA[77] with around 1 to 3 meV/atom (0.023 to 0.069 kcal/(mol · atom)), depending on the version (OMNI-P2x 0.11 kcal/(mol · atom) and AIMNet2[78] reporting a MAE of around 1.25 kcal/mol for neutral compounds. Although those results are not comparable one-to-one, as each of the models was trained and tested on a different dataset, they show that the ground-state error of OMNI-P2x remains within a reasonable range from current state-of-the-art machine-learning derived models, despite not being the main focus of this study. Correlation plots showing the predicted vs true excitation energies for all transitions are available as Supplementary Fig. 4. In all cases, the $R^2$ is above 0.91. For the ground-state energies in the test ANI-1ccx subset, OMNI-P2x has a higher MAE of 4.14 kcal/mol, but a high $R^2 = 0.9966$ for the normalized energies (0.9999 for unnormalized) (Fig. 4a). This is anticipated, as the ANI-1ccx

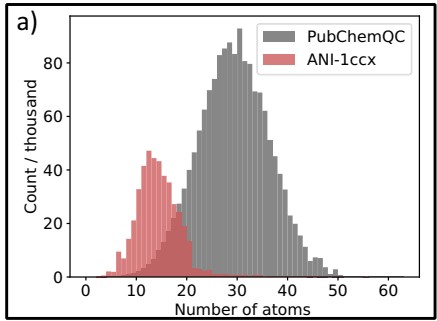
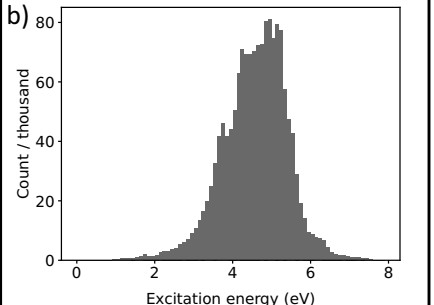

**Fig. 3 | Distribution of the data used to train OMNI-P2x.** Panel (**a**) shows a histogram of the distribution of the number of atoms in the molecules of the training sets PubChemQC (excited-state TD-DFT) and ANI-1ccx (ground-state CCSD(T)) (bin size 1); panel (**b**) shows the distribution of first excitation energies in PubChemQC (bin size 0.1 eV).

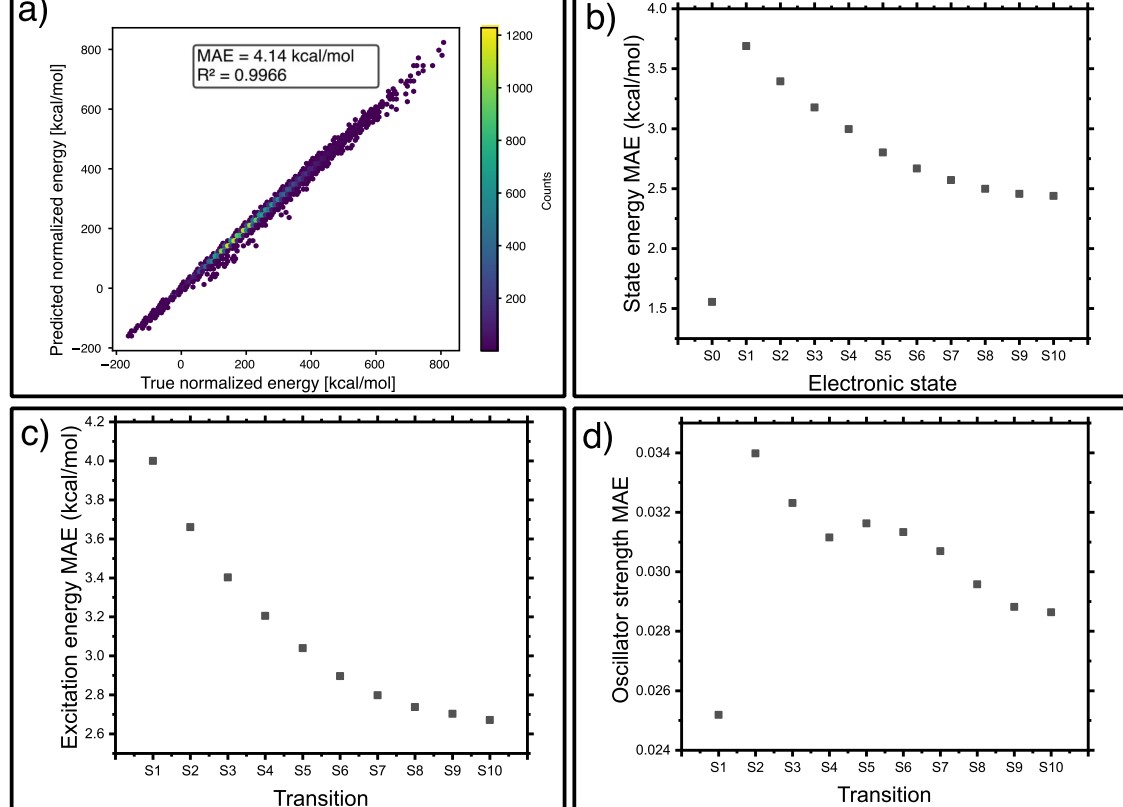

**Fig. 4 | Test performance of the OMNI-P2x model.** Panel (**a**): correlation plot of the reference (true) and OMNI-P2x predicted ground-state energies at the CCSD(T) level of theory, for the test part of the ANI-1ccx dataset; (**b**) Mean absolute error (MAE) of the electronic state energies. **c** MAE of the excitation energies. **d** MAE of the predicted oscillator strengths.

training data, which is only used as an auxiliary data set, is much smaller than PubChemQC, and is also featuring a different data distribution (Fig. 3a).

Additionally, OMNI-P2x provides an uncertainty quantification (UQ) of all predicted energies, using the standard deviations between quantities predicted by each of the three NN potentials as the error bar. A warning is issued to the user if the UQ of any excited states exceeds a safe threshold, which is determined as the median plus three median absolute deviations of the UQs of the test set, for each electronic state.

Finally, the MAE in OMNI-P2x predictions of oscillator strengths is below 0.04 for all studied electronic states (Fig. 4d). One can note that the highest accuracy in terms of the predicted oscillator strengths is achieved for the excitations to $S_1$ and to higher-lying $S_8$–$S_{10}$ states.

## UV/Vis absorption spectra

OMNI-P2x is a universal model that can predict both excitation energies and oscillator strengths, i.e., which allows one to obtain the UV/Vis excitation spectra of the QM quality, yet with a much higher speed (see Methods for details).

To test the OMNI-P2x performance for simulating the UV/Vis spectra, we generate the spectra for 50k test molecules from the PubChemQC database and compare them to the reference TD-DFT spectra (see Methods). Let us first examine representative spectra shown in Fig. 5. Looking at a spectrum belonging to the first quartile (top 25% spectra with the highest similarity between the OMNI-P2x and TD-DFT based on the similarity score, see Methods and below for details), agreement is observed both: in the intensity and position of

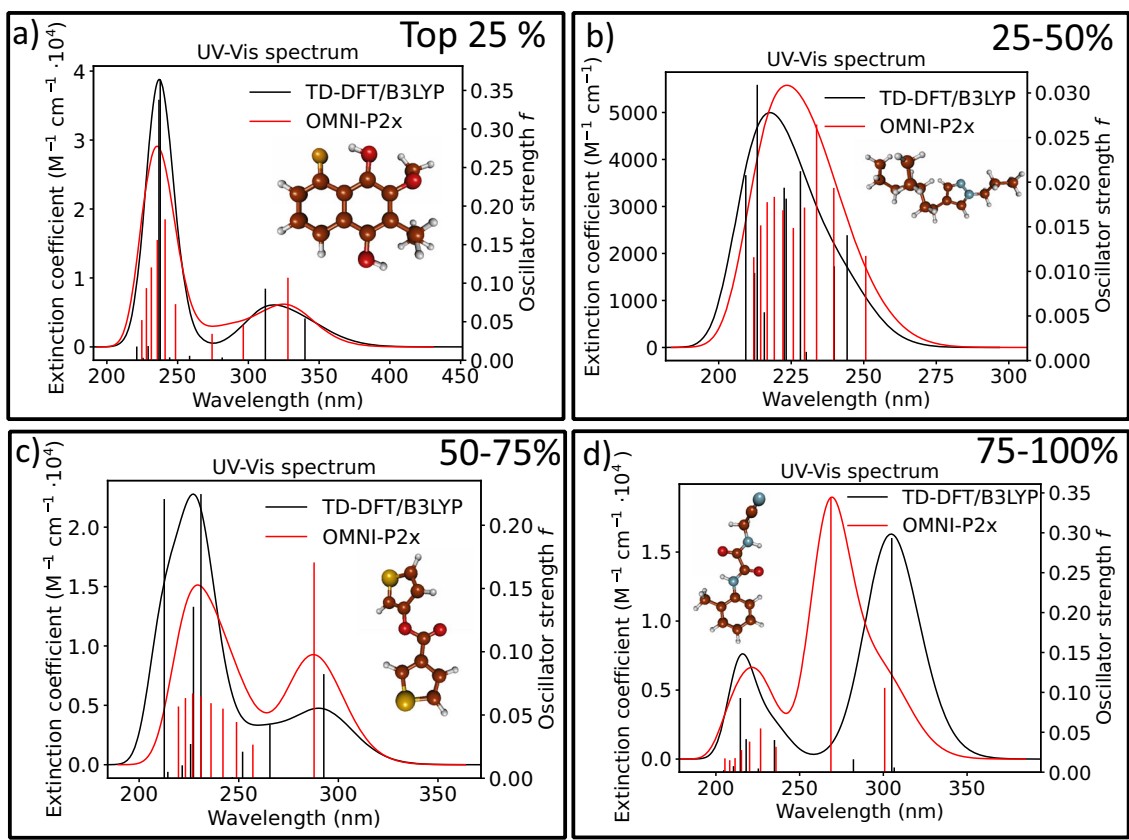

**Fig. 5 | Comparison between reference TD-DFT/B3LYP spectra and OMNI-P2x predicted spectra.** Four examples are shown, randomly chosen from the top 25% of the predicted spectra (panel **a**), ranked, 25–50% (panel **b**), 50–75% (panel **c**) and the bottom 25% (panel **d**).

the peaks (panel a in Fig. 5). For the second quartile, the agreement becomes somewhat worse, which is manifested by the absorption energy red-shift of about 10 nm, however, the intensity of the peak is consistent with the reference method (panel b). This agreement becomes worse in the bottom half of the predicted spectra: while the positions of the predicted peaks are rather accurate, the intensities are different from the reference method. Inspecting the bottom quartile of the predicted spectra, the agreement is still acceptable, with the higher absorption maximum predicted quite accurately, both in terms of the absorption energy and intensity, however, the first absorption peak appears to be blue-shifted due to the incorrect prediction of $S_2$ as a bright state. To provide a benchmark of the predicted spectra that is unaffected by the choice of a broadening factor in the SPC construction, we also rank the spectra by a geometric mean of the excitation energy and oscillator strength MAE, which is presented in Supplementary Fig. 5. Later, we also explore the application of OMNI-P2x for spectra calculations without relying on SPC approximation and the broadening factor.

For the test molecules shown in Figure 5, we tested the effect of finite-temperature distortions on the accuracy of OMNI-P2x and found that higher excited states remain close to equilibrium benchmarks, while the accuracy of the lowest excitations ($S_1$–$S_3$) may decrease more sharply. Violin plots showing the error distribution on non-equilibrium structures are available in Supplementary Figs. 6–9. To further probe this behavior, we performed relaxed PES scans of cyclohexanone (C=O stretch) and azobenzene (C–N=N–C torsion), shown in Suppl. Fig. 10. These scans confirm that OMNI-P2x is accurate near equilibrium and along moderate distortions, but errors grow for strongly compressed geometries or when torsion induces a change in excited-state character. Together, these results indicate that OMNI-P2x can extrapolate PES shapes close to equilibrium, but its reliability decreases as the molecular distortion drives the system beyond the domain of the training data.

To put the performance of the universal potential into perspective, we also evaluate two commonly used semiempirical QM methods: ODM2/CIS and TD-DFTB, on the same test of 50k molecules, as shown in Fig. 6. The OMNI-P2x spectra are clearly more similar to the reference TD-DFT than the semiempirical ODM2/CIS and TD-DFTB as judged by the distribution of the Spearman correlation coefficients for this test set (Fig. 6a). On average, the OMNI-P2x and TD-DFT spectra agree well with each other, with an average correlation coefficient of 0.86, and 80% of the predicted spectra having a correlation coefficient larger than 0.80. In comparison, both the semiempirical QM methods have a substantially worse performance, with an average SCC of 0.53 for TD-DFTB and 0.50 for ODM2/CIS. Additionally, ODM2's parameterization is limited to molecules consisting of H, C, N, O, F atoms only (39164 out of 50k structures), i.e., OMNI-P2x has a broader element coverage than ODM2/CIS. Noteworthy, the OMNI-P2x is not just more accurate than semiempirical QM methods, but is also clearly the fastest of the available approaches, with an average calculation time of 225 ms per predicted spectrum, while the ODM2/CIS and TD-DFTB approaches take on average 827 ms and 1152 ms, respectively (Fig. 6b). The reference TD-DFT calculations are orders of magnitude slower: more than 357.87 s per predicted spectrum. While the QM methods follow an approximately exponential scaling of cost with respect to accuracy, measured as correlation of the predicted spectra with TD-DFT/B3LYP spectra (note the logarithmic scale on the y axis), OMNI-P2x breaks this unfavorable scaling.

## Nuclear ensemble approach spectra

A more accurate method for predicting UV/Vis absorption spectra is the nuclear ensemble approach (NEA). In this technique, vertical excitation energies and oscillator strengths are computed for an ensemble of hundreds or thousands of molecular geometries[79,80]. The final spectrum is constructed by summing Lorentzian or Gaussian

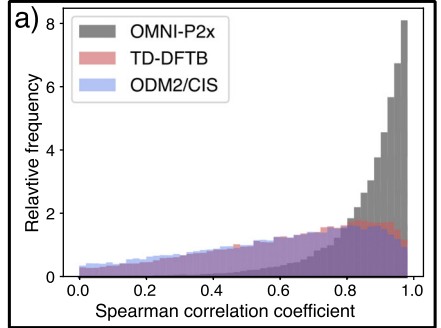
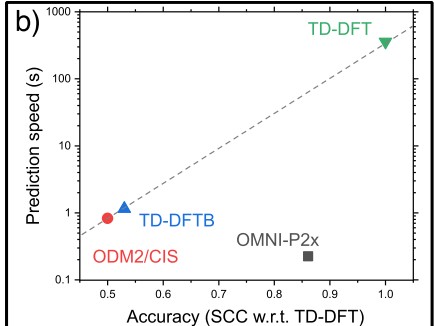

**Fig. 6 | The evaluation of the quality of the UV/Vis absorption spectra predicted by OMNI-P2x model on a test set of 50k molecules, at their ground-state equilibrium structures. a** The histogram of the correlation coefficients, quantifying the agreement between time-dependent density functional theory (TD-DFT) and OMNI-P2x electronic spectra, as well as two other commonly used cost-effective, semiempirical QM methods: orthogonalization and dispersion corrected method with configuration interaction single excitations (ODM2/CIS) and time-dependent density functional based tight binding (TD-DFTB). **b** The cost to accuracy correlation of OMNI-P2x, semiempirical QM, and TD-DFT spectra, with accuracy defined as the Spearman Correlation Coefficient (SCC) with respect to TD-DFT spectra, showing how the ML model breaks the traditional cost/accuracy tradeoff. The gray dashed line guides the eye to the difference in scaling between QM and ML methods.

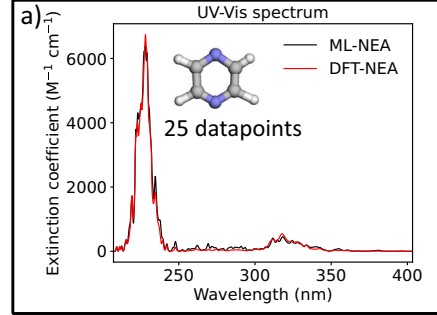
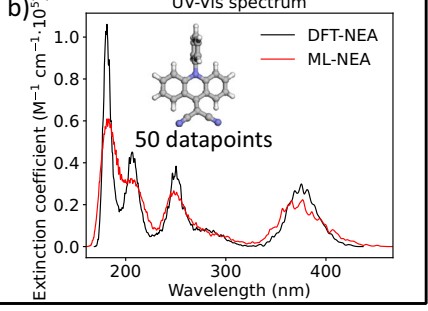

**Fig. 7 | Nuclear ensemble approach spectra simulations using OMNI-P2x.** The comparison between the ML-simulated (red) and TD-DFT (black) spectra for **a**) pyrazine and **b**) 9-DCMA.

functions centered at each geometry's excitation energy, weighted by the corresponding oscillator strength. Compared to single-point convolution, this approach better captures vibronic effects. However, NEA simulations are computationally demanding, as they require quantum mechanical calculations for all structures in the ensemble. To address this cost, ML models offer a natural path for acceleration[81–83]. Since OMNI-P2x is trained on excited-state data only from equilibrium structures, it is not directly applicable to NEA simulations. Nevertheless, fine-tuning enables it to achieve high data efficiency and simulation speed in this context, requiring a mere 25–50 additional, molecule-specific, training points to reproduce reference spectra.

To demonstrate this, we consider two test cases: pyrazine (with 4 excited states) and 9-dicyanomethylene derivative of acridine (9-DCMA, with 30 excited states), the latter of which was previously used to benchmark ML-NEA spectra with kernel ridge regression models[81] (see Methods for details). The NEA spectra simulated using reference TD-DFT data, as well as those predicted by the fine-tuned OMNI-P2x model−trained on just 25 data points for pyrazine and 50 for 9-DCMA−are shown in Fig. 7. For pyrazine (panel a), the agreement is observed across the entire spectrum. In the more complex case of 9-DCMA, the model accurately captures the intensity and position of the lower-lying excitations. Deviations start emerging below 250 nm, especially in predicted intensities, while the position of the peaks remains rather accurate. We attribute this to the model extrapolating in a regime beyond the number of excited states included in the training dataset: OMNI-P2x was pre-trained on 10 excited states, while 30 are considered in this example. Such simulations can be done on commodity hardware (Jupyter notebook using 4 CPUs), in under a

minute for pyrazine, and about 40 min for 9-DCMA, excluding the labeling time of QM methods.

## Interpertability of ML predictions

To provide interpretability of the ML-learned excitations, we introduce a new approach for analyzing atom-wise contributions to excitation energies. For each atom $i$, the fractional contribution is defined as $|\Delta E_i|/\Delta E$, where $\Delta E$ is the excitation energy for a given transition, and atomic contributions are calculated as the differences between ground- and excited-state atomic energies predicted by the NN ensemble. We show that the OMNI-P2x-derived atom-wise contributions agree well with the reference TD-DFT density difference plots (Fig. 8). Such analysis allows us to obtain the spatial character of the excitation by identifying which atoms are most involved in the transition. This analysis is conceptually related to work by Tkatchenko et al.[84], analyzing atom-wise contributions to ground-state energies.

We demonstrate that this interpretability analysis can work well across chemical space and for different excitations of the same molecule. Panel a) of Fig. 8 shows the $S_0 \rightarrow S_1$ ML $\Delta E$ contributions and TD-DFT density differences for several organic molecules. We begin by comparing the examples of benzene and pyrazine, two 6-$\pi$ electron aromatic compounds. OMNI-P2x correctly localizes the transitions, spreading it evenly across all the benzene carbon atoms, while in pyrazine it is mostly localized on the two nitrogen atoms, reproducing the TD-DFT results. The excitations also exhibit appropriate symmetry: $C_6$ for benzene, and $D_{2h}$ for pyrazine. The next example, azobenzene, shows physically correct localization of the excitation on the N=N double bond, while in $p$-nitroaniline the partial charge-transfer

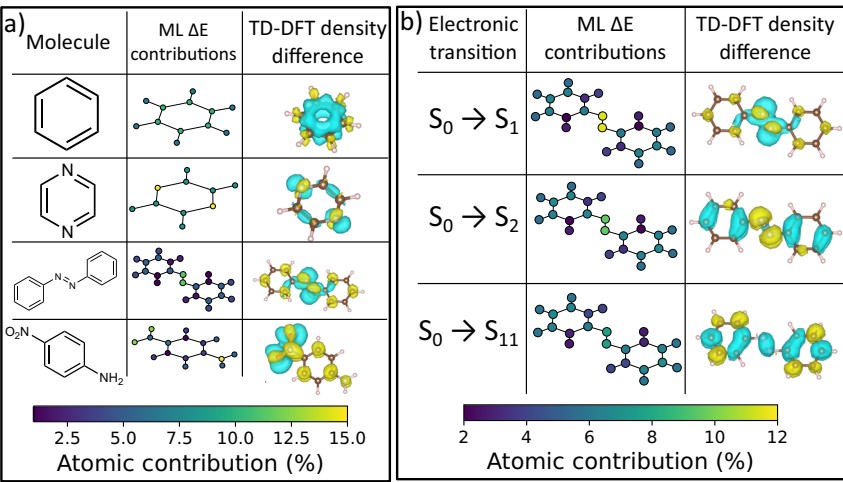

**Fig. 8 | Interpretability of OMNI-P2x excitation energies.** Panel (**a**) shows fractional atomic contributions to excitation energies for several different organic molecules, compared with time-dependent density functional theory (TD-DFT) density differences. Panel (**b**) shows different excited states of the same molecule, azobenzene.

character of the excited state is well reproduced. Overall, this shows an interpretable character of ML predictions, that is transferable across chemical space.

Atomic contributions to different electronic transitions of the same molecule are shown in Fig. 8b, using azobenzene as an example. OMNI-P2x correctly reproduces the $S_0 \rightarrow S_1$ transition as localized on the central N = N bond, while the $S_2$ transition has larger contributions from the aromatic rings. Finally, a higher excited state ($S_{11}$) is shown, which is delocalized across the entire conjugated system.

The means of analysis presented above do not exclude the need for dedicated analysis, either manual or using wavefunction-based tools such as TheoDORE[85] or Multiwfn[86], which can provide additional insights beyond spatial localization of the excitations, such as the orbitals' character. These tools are complementary, as massive-scale simulations enabled by OMNI-P2x can serve as a first step for generating enough statistical data, from where representative samples can be drawn, and post-processed further with the wavefunction-based tools.

### Design and screening of visible-light absorbing azobenzene compounds

The speed and accuracy of OMNI-P2x lend themselves for performing efficient high-throughput screening (HTS) of photo-active molecular systems. To demonstrate an application of OMNI-P2x for such a scenario, we perform the HTS of azobenzene derivatives characterizing with low-energy absorption. Our choice is motivated by the relevance of such compounds for applications in nonlinear optics[63], photo-responsive polymers[64], and as MOST systems[65,66]. To this end, firstly, we have generated a total of half a million azobenzene derivatives (Fig. 9) as SMILES strings, using RDkit[87]. In this procedure, we have allowed up to five substitutions with the following functional groups: methyl, ethyl, trifluoromethyl, nitro, hydroxyl, thiol, fluoro, chloro, amine, nitrile, and carboxyl. 3D structures (Cartesian coordinates) have been generated from SMILES using MLatom[88,89], and optimized using the fast and accurate AIQM3 model[76] (see Methods for details). Subsequently, we have predicted the excitation energies of the optimized structures with OMNI-P2x. To refine the quality of the OMNI-P2x predictions, 1000 random structures have been sampled from the generated set and labeled using the TD-DFT/B3LYP method (see Methods for details on the level of theory). We have then fine-tuned two copies of one of the OMNI-P2x energy models on this set, using different training/validations splits. The procedure of prediction, selection, and optimization has been repeated two more times, at each

iteration adding the 1000 most uncertain molecules to the training set, with the uncertainty measured by the deviation between the two trained models. When a correlation coefficient of 0.78 and an MAE of 0.25 eV have been reached as tested on 1000 randomly sampled structures, we have stopped the first phase of the screening procedure, as this accuracy is similar to the accuracy of the target TD-DFT methods. For the 1000 lowest-energy absorbing candidates, as predicted by the fine-tuned model, TD-DFT predictions have been made again. Out of these structures, a total of 21 lead candidates with transition energies below 2.0 eV have been determined (see Supporting Material, SM). Due to the potential charge-transfer character of excitations in these molecules, which the B3LYP functional would be unable to capture correctly, we have used at this stage a higher-level electronic structure method, ADC(2)[16], to re-evaluate the excitation energies of the 21 lead candidates.

The three most red-shifted derivatives identified with the described procedure are presented at the bottom of Fig. 9, while a database containing all the structures and their computed UV/Vis spectra is available in the SM. A plot comparing the spectra predicted at the three studied levels of theory is available in Supplementary Note 3. The predicted absorption energies of the top three derivatives are lower than the absorption energies of reported human-designed red-shifted azobenzene units with the same core structure[90–93]. At the same time, our top candidates determined via in silico screening show similarities to substitution patterns of the human-designed derivatives, i.e., ortho- and para-placements of electron-donating groups[94,95].

### Efficient ML-driven nonadiabatic dynamics

As a universal model, OMNI-P2x has the potential to accelerate non-adiabatic dynamics (NAMD) simulations, similarly to how ground-state models are used to efficiently perform ground-state dynamics. However, in this respect, it should be noted that NAMD is not just much more computationally demanding than the ground-state dynamics, but also typically requires a system-specific selection of a level of theory, as the common black-box approaches, like TD-DFT, cannot properly describe the key features of excited-state PES, i.e., the first excited to ground-state PES intersection[96]. This often necessitates a suitable choice of more advanced QM methods, such as CASSCF or CASPT2, which require the active spaces specifically selected for a given system. Since our OMNI-P2x is only trained on TD-DFT data, here, instead of directly using it to propagate NAMD, we consider a more realistic scenario, in which we want to fine-tune OMNI-P2x for the

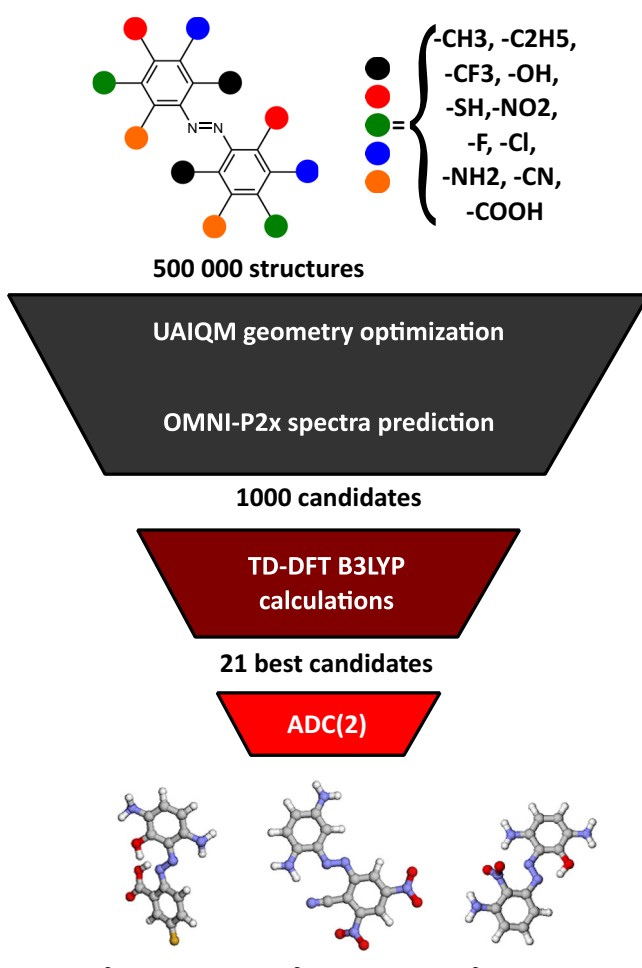

**500 000 structures**

UAIQM geometry optimization

OMNI-P2x spectra prediction

**1000 candidates**

TD-DFT B3LYP calculations

**21 best candidates**

ADC(2)

λ = 667 nm λ = 687 nm λ = 640 nm

**Fig. 9 | Summary of the high-throughput screening process for red-shifted, visible-light absorbing azobenzene derivatives.** Out of the 500 k generated structures, iterative fine-tuning of the OMNI-P2x model on structures optimized with UAIQM (see "Methods") determined 1000 candidates for which time-dependent density functional theory (TD-DFT) spectra were computed. The procedure was repeated several times. From these structures, higher-quality spectra using the second-order algebraic diagrammatic construction (ADC(2)) method were computed for the best candidates, yielding 21 compact, visible light-absorbing azobenzene derivatives, with the three most red-shifted compounds presented in the figure.

specific problem, i.e., the given combination of the desired QM level and system.

To perform the fine-tuning as data-efficient, as possible, we generate data to fine-tune the pre-trained universal potential OMNI-P2x by modifying our previously developed active learning procedure[56], where we use the OMNI-P2x as a starting point in each iteration of the active learning loop. Performing direct active learning from scratch, i.e., generating data for a neural network potential with random weights, has been, so far, state of the art in terms of time- and data-efficiency for ML-driven NAMD, but it might still take several days to weeks of wall-clock time to converge, even for relatively simple systems, such as fulvene or azobenzene[56].

As our first test system, we use fulvene, popularly chosen as a prototypical molecule for benchmarking NAMD methods[97]. This choice also allows us to compare our new protocol, based on fine-tuning OMNI-P2x, to the previous state-of-the-art protocol using MS-ANI with random initial weights[56]. As the reference electronic-structure method, we use CASSCF (6,6). The NAMD is performed with the Landau–Zener–Belyaev–Lebedev (LZBL) variant of TSH[98–102], which

does not require the calculation of the nonadiabatic coupling vectors (see Methods). With active transfer learning employing fine-tuning of the universal potential OMNI-P2x in the active learning loop, we are able to slash the convergence time of fulvene from over three days to about 24 h, with a ca. three-fold increase in computational efficiency. Active transfer learning yielded only 3550 conformations, which is about twice fewer than the number of training points required to achieve convergence in active learning procedure performed from scratch, as can be seen in Fig. 10a. Moreover, inspecting the time evolution of the electronic state populations, we can see a statistically significant difference in the populations predicted by the fine-tuned model and the previous MS-ANI model. Despite a smaller training set, the TL version of OMNI-P2x better captures the photophysical behavior of the system. A comparison of the dynamics predicted using different ML models and different LZBL propagation settings with reference fewest switches surface hopping dynamics is available in Supplementary Note 4. Further analysis of the NAMD trajectories of fulvene, including comparison of the ML-predicted potential energy surfaces, minimum energy conical intersections, as well as hopping points and deactivation channels, is provided in Supplementary Note 4.1. In all cases, agreement between the ML model and reference CASSCF calculations is noted.

This improvement is even more substantial when moving to more complex cases of active learning. One such example is the photo-isomerization reaction of azobenzene, simulated at the AIQM1/MRCI-SD level of theory (see Methods). Due to a large volume of chemical space covered during such a process, learning photoreactions is a particularly difficult task. To assess the improvement in data-efficiency that comes from using a pre-trained universal model, we compare the performance of the OMNI-P2x fine-tuned and MS-ANI trained from scratch on fractions of the full training set required to converge an AL procedure (35071 points, as reported in ref. 56). Looking at the stability of the dynamics, calculated as the percentage of trajectories finishing without dissociation, shown in Fig. 10b, the fine-tuned OMNI-P2x model can propagate stable dynamics given just 0.1% of the dataset, while dynamics propagated using the pure learning approach become consistently stable above 10% of the data. Furthermore, Fig. 10c presents the predicted quantum yield of *trans* to *cis* photoisomerization, as a function of the training set size. For the TL potentials, all models trained on above 2% of the data predict QYs within the reference range, up to a 95% confidence interval error bar. On the other hand, the performance of the models trained from scratch is much less consistent, with outliers predicting incorrect quantum yields even at high fractions of the training set. An analysis of azobenzene $S_0$ and $S_1$ PES', optimized MECIs, and deactivation geometries is provided in Supplementary Note 4.2, along with a comparison to reference results, which agree with the ML models.

The above findings highlight that pre-training of OMNI-P2x on a vast dataset allows it to gain knowledge on both the chemical compound space and some information about conformational space, greatly reducing the amount of data needed for its fine-tuning. This makes OMNI-P2x a method of choice for propagating ML-accelerated surface hopping dynamics after data-efficient fine-tuning. A similar phenomenon has been reported in refs. 57,103, where a neural network potential pre-trained on ground-state data could achieve a lower MAE for excited-state predictions than a potential trained from scratch, in the low-data regime.

## Discussion

We present OMNI-P2x: a universal ML potential for predicting key molecular excited-state properties, including energies, forces, and oscillator strengths. The model enables out-of-the-box predictions of electronic spectra of organic molecules, targeting the TD-DFT/B3LYP level of theory, with an accuracy of about 0.15 eV. Importantly, OMNI-P2x offers higher-quality spectra predictions than commonly used

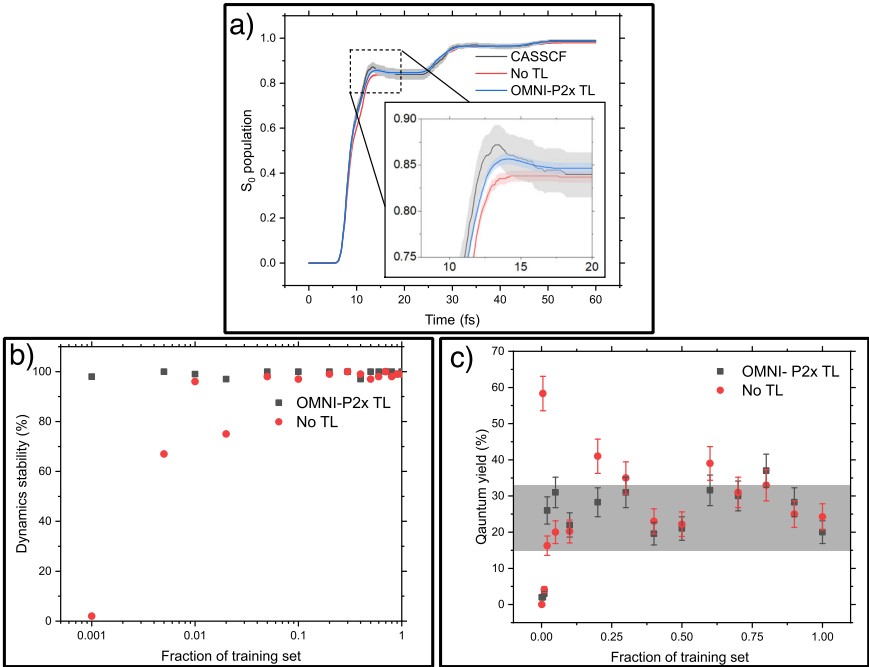

**Fig. 10 | Performance of fine-tuned OMNI-P2x in NAMD simulations.** Panel (**a**) shows the ground-state population evolution of fulvene, from NAMD trajectories propagated using ML models trained in an active learning loop from scratch (red line) and by fine-tuning OMNI-P2x (blue line), as well as reference complete active space self-consistent field method (CASSCF) dynamics (black line), along with shaded, 95% confidence interval error bars. Panel (**b**) shows the stability of the NAMD simulations of azobenzene photoswitching as a function of the full active learning dataset, while panel (**c**) shows the predicted photoisomerisation quantum yield by models trained on increasing fractions of the training set, with a shaded area corresponding to the prediction of the reference method, from ref. 56. TL -- transfer learning. Error bars were estimated using the normal approximation interval for a binomial process with a confidence interval of 95%.

semiempirical QM methods for excited states, while having a substantially lower computational cost. This efficiency and accuracy in electronic spectra predictions enable high-throughput screening of photo-active molecular systems with required properties: we demonstrate this by screening half a million of azobenzene derivatives for red-shifted, visible light-absorbing photoswitches. The screening delivered three top candidates absorbing in longer-wavelength regions than previously reported azobenzene derivatives.

As with all machine-learning interatomic potentials, OMNI-P2x can only be as accurate as the reference method on which it is trained. In our case, this is TD-DFT/B3LYP/6-31+G*, which offers a practical balance between accuracy and computational feasibility across large chemical spaces, but is known to struggle with certain excited-state properties—most notably charge-transfer and Rydberg excitations. In small molecules, the use of a modest basis set such as 6-31 + G* can further exacerbate this issue, as higher-lying excitations may correspond to poorly described Rydberg states. As such, the expected level of error of TD-DFT/B3LYP, which is usually between 0.2 and 0.4 eV with respect to experimental benchmarks[104,105], or theoretical best estimates (TBE)[106–109], will be retained. The same is true for the oscillator strengths, which have reported MAE of between 0.05 and 0.15, compared to TBE[106,110]. We supplement these benchmarks by providing a comparison between ADC(2) and TD-DFT/B3LYP vertical excitation energies in Suppl. Fig. 19 and Supplementary Table 3. The DFT results correlate well with ADC(2), showing a systematic underestimation of excitation energies, that increases when moving to higher excited states. These limitations are intrinsic to the reference data and thus propagate into the trained model. Furthermore, the neural network model is expected to produce reliable predictions primarily for molecules whose size and chemical composition are similar to those represented in the training set; extrapolation to significantly larger or chemically distinct molecules may lead to unphysical or unexpected results. Nevertheless, OMNI-P2x is fully agnostic to the level of theory

when fine-tuned on more accurate data when available; in fact, we show in this work that transfer learning from TD-DFT/B3LYP already improves performance when adapted to higher-level reference methods.

Importantly, we show that the OMNI-P2x universal model can be used to accelerate and increase the robustness of the ML-driven NAMD simulations. For this, we developed a protocol using fine-tuning OMNI-P2x in the active learning loop, allowing up to a ten-fold increase in the data efficiency compared to active learning from scratch. The quality of the resulting ML-driven NAMD is retained or even improved compared to the ML-NAMD without using OMNI-P2x. The pre-trained weights in OMNI-P2x contain the knowledge of the vast chemical compound and conformational space, which imparts ML-driven NAMD greater stability compared to the ML dynamics performed with the networks trained from scratch. This advantage has already been used in the follow-up study, enabling the NAMD simulations with electronic-structure methods without analytical gradients such as QD-NEVPT2[111].

While OMNI-P2x is the first universal excited-state neural network potential, it naturally has scope for further enhancements. Expanding the training set could enable support for a wider range of element types and out-of-equilibrium structures. Future extensions of the framework will aim to predict not only for neutral molecules in the singlet manifold, but also for states of higher multiplicity and charged species. Although the present focus on singlet states already enables accurate and chemically diverse predictions for many photophysical applications, the architecture is inherently extensible, and these additional capabilities will be the subject of future work.

Additionally, developing a universal potential capable of performing NAMD simulations across chemical space without the need of fine-tuning remains an arduous task due to the diversity of photochemical transformations, high variations in molecular conformations in the direct vicinity of conical intersections, many possible choices of QM levels and settings inside the QM levels (e.g., active space), and

would require extensive training data and, possibly, additional modifications to the architecture. In the meantime, the protocol combining fine-tuning of the OMNI-P2x with active learning, as presented here, can substantially ameliorate this problem by providing specific solutions for the problem in hand.

# Methods

## Model training and data collection

The entire PubChemQC dataset[69] was obtained from the PubChemQC Project (https://nakatamaho.riken.jp/pubchemqc.riken.jp/b3lyp_2017.html). The molecules in the PubChemQC dataset were optimized at the level of B3LYP[71]/6-31G*[70], and excited-state properties were calculated with TD-B3LYP/6-31+G*[70] using GAMESS[112]. For each molecule, the first 11 electronic states were calculated, leading to 10 excitation energies and oscillator strengths.

In this study, the molecules were selected from the PubChemQC dataset, which satisfied the following conditions: multiplicity of 1, net charge of 0, and TD-DFT data for 11 electronic states, and containing elements C, H, O, N, F, S, and Cl. The resulting data set contained 3.1 million molecules, which we compiled with MLatom 3.16.2[88,89,102] to MLatom's native machine-readable database format and saved in JSON files, used for training and testing in this study.

During training, the MSE loss function for energies-only was used:

$$L = \left( E^{\mathrm{ML}} - E^{\mathrm{ref}} \right)^2, \tag{1}$$

where $E^{\mathrm{ML}}$ and $E^{\mathrm{ref}}$ are the ML-predicted and reference energies, respectively.

Analogously, the MSE loss for oscillator strengths was used for a model trained on oscillator strengths.

## UV/Vis spectra prediction

The electronic, UV/Vis absorption spectra are obtained using the standard single-point convolution approximation based on the broadening function as derived in ref. 113:

$$L(x) = 0.619 \sum_i \frac{f_i}{\Gamma} e^{-(x-x_i)^2/\Gamma^2}, \tag{2}$$

where $x_i$ are the predicted transition energies, $f_i$ are their oscillator strengths, and $\Gamma$ is a pre-defined full-width at half maximum parameter (FWHM). In this work, a broadening parameter of 0.3 eV was used, as recommended in ref. 113.

Electronic spectra predicted by different methods were compared using Spearman's rank correlation coefficient (SCC), $r_{\mathrm{SCC}}$:

$$r_{\mathrm{SCC}} = 1 - \frac{1}{n(n^2-1)} \sum_i d_i^2, \tag{3}$$

where $d_i$ is the difference in rank between the two spectra, and $n$ is the total length of the spectrum. The ODM2/CIS spectra were computed using the MLatom's interface to the MNDO package[114], and TD-DFTB spectra were obtained using an interface to the DFTB+ program, release 24.1,[115] with the 3ob-3-1 set of parameters[116].

To address non-covalently bound structures separated by distances larger than the molecular descriptor cutoff, we introduce a fragment correction scheme (Supplementary Fig. 1). The method checks for non-covalently bound fragments and compares excitation energies of the full system against those of individual fragments to determine whether the state corresponds to localized excitations or an excimer. Validation on a benzene dimer as a function of intermolecular separation shows that the corrected model agrees well with TD-DFT/B3LYP reference values, whereas the uncorrected model diverges

when parts of the system fall outside the cutoff radius. Full details are provided in Supplementary Note 1.

For the nuclear ensemble approach spectra, 500 conformations of pyrazine were sampled from a 298 K Wigner distribution, as implemented in MLatom, and labeled using TD-DFT/B3LYP (6-31+G*) in Gaussian 16.0 (Revision C.01) software package[117]. Geometry optimization was performed using the geomeTRIC[118] package, and PySCF[119] was used for frequency calculations. Data for the 9-DCMA system was extracted from ref. 81, where it was computed at the CAM-B3LYP[120]/ma-TZVP[121] level of theory.

The thermally distorted structures of molecules presented in Supplementary Figs. S6–S9 were sampled from a 2-ps-long ground-state Born–Oppenheimer molecular dynamics trajectory, with the first 1 ps discarded for thermalization, performed at the GFN2-xTB level of theory[122–124]. All the calculations and plots were prepared with MLatom.

## NAMD simulations

CASSCF calculations used as a reference method for TSH simulations were performed through the interface to the COLUMBUS quantum chemistry package[125], using an active space of 6 electrons in 6 orbitals. TSH simulations were performed using the Landau–Zener–Belyaev–Lebedev formalism[98–102], with hopping probabilities between states k and j, $P_{j \to k}$ calculated using the following formula:

$$P_{j \to k} = \exp \left( \frac{-\pi}{2\hbar} \sqrt{\frac{Z_{jk}^3}{\ddot{Z}_{jk}}} \right), \tag{4}$$

where $Z_{jk}$ is the energy gap between states j and k, and $\ddot{Z}_{jk}$ is its second-order time derivative.

The active learning loop uses the standard settings presented in ref. 56, with initial points sampled from a harmonic approximation Wigner distribution, for a total of 250 points. The maximum propagation time was set to 60 fs with a time step of 0.1 fs. Velocities after hopping were rescaled in the direction of the momentum vector, using a reduced kinetic energy reservoir[126]. In each iteration of the AL procedure, 50 ML-TSH trajectories were run, and 50 gapMD trajectories. 15 additional points were sampled based on hopping probability uncertainty, up to a threshold of 300 points.

Testing of the model was performed using 10000 trajectories for the MS-ANI model, as well as the fine-tuned OMNI-P2x model, and 623 CASSCF(6,6) trajectories were used as reference. One model from the energy ensemble of OMNI-P2x was used for fine-tuning in all NAMD applications. For both the studied systems, inital conditions were generated by sampling from a 298K Wigner distribution, as implemented in MLatom, using the same software for optimization and frequencies as for pyrazine.

The reference data for azobenzene dynamics was labeled using the AIQM1/MRCI-SD method[127]. In the semiempirical part of AIQM1, the half-electron restricted open-shell Hartree–Fock formalism[128] was used in the SCF step, with the singly-occupied HOMO and LUMO orbitals. Two additional closed-shell references were added in the MRCI procedure, a HOMO–HOMO configuration and a doubly excited, LUMO–LUMO configuration. The active space consisted of 8 electrons in 10 orbitals (four occupied orbitals, six unoccupied). Single and double excitations within a such-defined active space were allowed. 100 trajectories were ran and analyzed for both the *trans → cis* and *cis → trans* photoisomerization reaction. The same number of trajectories was used when testing the model that was trained from scratch.

The simulated quantum yields Φ were calculated as the fraction of the trajectories relaxing to form the photoproduct (the other isomer), $N_{\mathrm{reactive}}$, to the total number of trajectories $N_{\mathrm{traj}}$: $\Phi = N_{\mathrm{reactive}}/N_{\mathrm{traj}}$.

## Azobenzene screening

All computations were performed with the XACS version of MLatom (now MLatom with Aitomic add-ons)[102,129,130]. We used the MLatom interface to the TorchANI[68] package for all models based on ANI-type networks and their modifications (i.e., MS-ANI, OMNI-P2x, and AIQM3[76]). The AIQM3 calculations also used the MLatom interface to the s-dftd3 program[131], providing the D3(BJ) corrections[132] and the xtb program[133] providing the GFN2-xTB* baseline (GFN2-xTB[122–124] without the D4 dispersion corrections, e.g., as in the AIQM2 method[75]). The azobenzene derivatives were optimized with the AIQM3 version of the UAIQM models[72] targeting the CCSD(T)/CBS accuracy; this model is available as an add-on to MLatom provided via the Aitomic package[134]. For TD-DFT calculations, MLatom's interface to the Gaussian 16 software package (Revision C.01)[117] was used. ADC(2)-level spectra were computed using the Turbomole[135] software package, with the Dunning correlation-consistent double-$\zeta$ basis set (cc-pVDZ)[136].

## Data availability

The PubChemQC and ANI-1ccx datasets are publicly available at https://doi.org/10.1021/acs.jcim.7b00083 and https://doi.org/10.1038/s41597-020-0473-z, respectively. Data supporting the findings, including the screening of azobenzene derivatives and nonadiabatic molecular dynamics trajectories, are deposited on Figshare, at https://doi.org/10.6084/m9.figshare.28794599. Source data are provided in this paper.

## Code availability

The code is available in the open-source MLatom under the MIT license as described at https://github.com/dralgroup/mlatom with the OMNI-P2x model weights and scripts relevant to this publication available in a dedicated repository under the MIT license at https://github.com/dralgroup/omni-p2x. The calculations can also be performed online on Aitomistic Lab@XMU (https://atom.xmu.edu.cn) and Aitomistic Hub (https://aitomistic.xyz) with convenient GUIs and Aitomia (or its successors such as Protomia) – AI agents assisting in simulations via natural language prompts[137].

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

## Acknowledgements

M.M. acknowledges the Polish Ministry of Education and Science for funding this research under the program "Perły Nauki," grant number PN/01/0064/2022, amount of funding and the total value of the project: 239 800,00 PLN, as well as gratefully acknowledge Polish high-performance computing infrastructure PLGrid (HPC Centers: ACK Cyfronet AGH) for providing computer facilities and support within computational grant no. PLG/2024/017363. P.O.D. acknowledges funding by the projects for International Senior Scientists (Project No.: W2531013) and for Outstanding Youth Scholars (Overseas, 2021) of the National Natural Science Foundation of China, via the Lab project of the State Key Laboratory of Physical Chemistry of Solid Surfaces, and by Aitomistic, Shenzhen. Part of the computations were performed on the Xiamen Atomistic Computing Suite cloud server at http://XACScloud.com (succeeded by Aitomistic Lab@XMU at https://atom.xmu.edu.cn). The authors also thank Yuxinxin Chen for providing the AIQM3 method of better quality for optimizing geometries of the azobenzene derivatives, before this method publication.

## Author contributions

M.M. developed the code for training the OMNI-P2x and MS-ANI models, tested the collected datasets, performed model training, benchmarking and testing, performed the fine-tuning experiments, designed the screening pipeline and performed the required calculations and model training, developed the modified active learning protocol for TSH and performed the NAMD simulations, analyzed the results, wrote the original version of the manuscript and prepared the figures. X.Y.T. performed data collection and initial evaluation of their quality via MS-ANI model training, implemented the interface to DFTB + (used for TD-DFTB calculations), and assisted in optimization of azobenzene derivatives. J.J. participated in the result analysis and interpretation, co-supervised research, and assisted in the writing of the original version of the manuscript. P.O.D. designed and conceived the project, contributed to the result analysis and interpretation, co-supervised research, wrote the initial manuscript's outline, title and abstract, and secured funding. All authors discussed the results, contributed to the manuscript and its revisions.

## Competing interests

The authors declare no competing interests.
