## [Transparent Peer Review file · Nature Communications]

OMNI-P2x Universal Neural Network Potential for Excited-State Simulations

Corresponding Author: Professor Pavlo Dral

Version 0:

Reviewer comments:

Reviewer #1

(Remarks to the Author)

The manuscript introduces OMNI-P2x as a universal neural network potential for molecular excited and ground state energy prediction. The model is inspired by the multi-state ANI model, based on own prior work by the authors [54], and as a novelty in this manuscript, adds all-in-one learning to simultaneously learn from multiple datasets with different levels of theory.

The manuscript is a step forward towards generalizable prediction of excited states energies, but there are a few questions that should be discussed before publication, in order to not only emphasize the strengths but also the limitations of this work.

Page 6: "The predictions are then made atom-wise and summed up to yield the total energy of the molecule."  Does that mean that also the excited states energies are predicted by a sum of atom contributions? How are the labels for the excited states pre-processed (relative to the total energy of the ground state or as absolute energies), and how are they normalized?

A simple test with your code and a duplicated and shifted benzene molecule shows that the excitation energies double when duplicating (and shifting) the input molecular structure. Of course, the model was not trained on such an input system, but the inductive bias given by the summation of single-atom contributions is not suitable for the prediction of excitation energies as non-extensive quantities (in contrast to ground state energies):

```
import mlatom as ml
omni = ml.models.methods(method='omni-p2x')
```

```
mol = ml.data.molecule()
mol.read_from_xyz_file("Benzene.xyz")
omni.predict(molecule=mol, nstates=5)
mol.excitation_energies
0.18915822, 0.20142478, 0.2106019, 0.21771565
```

```
mol2 = ml.data.molecule()
mol2.read_from_xyz_file("Benzene2.xyz") # two benzenes, the second is shifted by 10A in +z direction.
omni.predict(molecule=mol2, nstates=5)
mol2.excitation_energies:
0.37831637, 0.40284953, 0.42120379, 0.4354313 # roughly twice the excitation energies of the single benzene molecule.
```

Figure 2: How exactly is the state and the level of theory encoded? Figure 2 suggests that the state is an integer number appended to the molecular geometry vector, while the level of theory is represented as multiple one-hot encodings of length two (which seems not intuitive). Why is the state not also represented as a one-hot encoding? Do you have empirical evidence that this is the most informative / data-efficient representation?

Page 7: "PubChemQC project's dataset consists of unique, optimized structures of molecules at the TD-B3LYP/6-31+G* level of theory, with the energies and oscillator strengths of the first 10 excited states calculated using TD-DFT."  How does the model without transfer learning perform on excited states of equilibrium and non-equilibrium structures? Does the model without transfer learning just apply the learned ground state PES also to the excited states, so does it overestimate

the correlation between ground and excited states, i.e. parallel ground- and excited states surfaces? Or does it somehow learn to predict excited states PES (at least close to equilibrium structures), even though it was only trained on excited states energies at equilibrium structures?

Figure 4a shows total energies, but it is unclear if those are ground-state or excited-state total energies. Furthermore, calculating the r^2 score on non-normalized total energies is misleading. Even just applying the atom-specific normalization to total energies without learning anything will lead to very high r^2 scores. The r^2 score should be calculated on the predictions on the normalized scale, not on the unnormalized scale.

Figure 4 shows three different units. It would be better to discuss everything in one unit (e.g. eV), and then additionally convert key quantities to kcal/mol. But showing energies in Hartree in panel (a), MAEs in kcal/mol in (a) and (b), and then MAEs in eV in panel (c) is very confusing. Additionally, the difference between (b) and (c) does not become fully clear from the caption or text.

Excitation energy predictions (relative to the ground state, separated by state) should be plotted in scatter plots, and r^2 values should be calculated. Figure 3b indicates a standard deviation of ~ 1 eV for excitation energies in the dataset, but I assume that the std is smaller if this is separated by state. Figure 4c indicates MAEs of ~ 0.2 eV, which is significant compared to the variance of the labels. Therefore, I assume that the r^2 scores are not 0.9999, as calculated for the non-normalized ground state energies. A more open discussion about the actual accuracy (also in terms of r^2 scores) is absolutely necessary to judge the reliability of the results and the impact of this study.

Figure 6a/b fully focuses on equilibrium structures, which makes sense given the training data of the OMNI-P2x model, but which is not a fully fair comparison. Therefore, this should be explicitly mentioned in the caption and in the text/discussion. If possible, non-equilibrium structures of the same molecules should be sampled at a given temperature to see how fast the OMNI-P2x accuracy decays when going away from equilibrium geometries.

Figure 6b suggests that ODM2/CIS and TD-DFTB are not on the Pareto front of speed and accuracy. However, it would be interesting to see how a delta learning approach that combines ODM2/CIS or TD-DFTB with OMNI-P2x would perform. The cost is roughly the same as the semi-empirical methods (because OMNI-P2x inference time is negligible), but the overall accuracy might be above the accuracy of OMNI-P2x.

All lines in Panel 8a seem more or less identical. It is not clear if the TL approach is statistically significantly better than the No-TL approach. A zoom into a particular part of the plot and the addition of uncertainty intervals is necessary to support the claims. Furthermore, the accuracy of TL and No-TP approaches in the population-time-curve should be quantified using an appropriate metric and plotted as a function of the active learning dataset size to obtain more insight into the advantages of TL here. The examples from 8c/d are not enough.

(Remarks on code availability)

I reviewed and tested the code. Everything works, the Jupyter notebook is reproducible. However, the library is not necessarily organised as you would expect from a professional software, which might not be necessary in this context.

Reviewer #2

(Remarks to the Author)

In their manuscript, Martyka et al. report the development of two artificial neural networks that, combined, predict approximate TD-DFT excitation energies and oscillator strengths of molecules. This is obtained by combining existing and established architectures by including data at CCSD(T)/CBS and TD-DFT B3LYP/6-31+G* levels of theory. The authors test their model on three applications, namely the convolution of absorption spectra based on predicted energies and oscillator strengths, the rapid screening of azobenzene derivatives to discover new candidates with desired properties, and the use of the model as a starting point to further fine-tune it with active learning to compute energies and gradients for nonadiabatic dynamics. While the manuscript introduces a potentially interesting architecture that contributes to increasing the potential of and reducing the limitations of artificial intelligence in the field of computational photochemistry, this work appears to have some theoretical limits and rather limited impact, breakthrough, and interest to justify publication in Nature Communications. Additionally, the authors claim the universality of the model, which would mislead the wide readership of this journal that might interpret the model as actually universal and flawless, which, in my opinion, it is far from.

There are three reasons why I disagree with the universality and the high impact of the model:

- The model seems definitely transferable (with debatable prediction accuracy) among the chemical space similar to the one used for the training. However, this is limited to a certain type of molecule (only containing H, C, N, O, F, S, and Cl atoms), leaving out large categories of important photoactive molecules. For example, no transition metal complexes are considered, whose inclusion would make the model more universal.
- There is no information about the character of the states, triplet states energies, nonadiabatic and spin-orbit coupling are not predicted, and there is no information about how to deal with ionic molecules. The model is able to predict vertical singlet excitations with a feature for the state order, but there is still no information about the orbitals involved and the actual character of the excited states. The universality is claimed to be in terms of how many different molecules can be predicted, but not in terms of photophysical properties and accuracy.
- The training on the TD-DFT data makes for sure possible to predict energies and oscillator strength, but this is still training on wrong results in most cases, as the functional chosen is well known to have very poor reliability for the description of fundamental features of excited states. So a universal model should predict good and physically meaningful results, and not have as a goal to perform better than poorly reliable semi-empirical approaches.

Additionally, I found other major issues with the manuscript:

1. The only task for which I would recommend a trustworthy use of the model is the virtual screening of azobenzene derivatives, where being able to qualitatively and rapidly screen molecules can help identify and select potential candidates with desirable properties. Sadly, the quantitative performance shown in the UV/Vis test case is very poor. Even in the (potentially cherry-picked) example from the top quartile, the prediction is, at least partially, wrong. The oscillator strengths are off, and between 275 and 300 nm there are 4 states predicted vs 1 at TD-DFT/BLYP level. The authors claim an excellent agreement, probably referring to the position of the maxima, but anyway, the convolution of the spectra is highly affected by the artificial broadening of 0.3 eV, which might make the results look more in agreement than they actually are (additionally, the reference to justify 0.3 is unrelated, as in this manuscript the error is between TD-DFT data and their prediction, and not a discrepancy between computed/computed at different levels or computed/experimental spectra)
2. The overall idea of the UV/Vis task is to predict the brightness of states in a specific order. This is not enough for this task; it is just a collection of energies with no correlation with their brightness. No information about the character of the states is present, and why a state is bright or not. The adiabatic order of the states (S_1 , S_2 , etc.) has no correlation with what states they actually are. Two molecules could have S_4 both bright for two completely different and uncorrelated reasons, while two very similar molecules could have S_1 and S_2 with inverted brightness just because, for example, $\text{n}\pi^*$ and $\text{p}\pi^*$ states are inverted. So if there is no information on the mapping between excited states, predicting oscillator strength of adiabatic states is useless, if not wrong.
3. In this logic, I also cannot understand, and it's not explained, why the highest accuracy is achieved for the S_0 excitations to S_1 and to higher-lying S_8 – S_{10} states. I cannot see any correlation in that, as there is no information on the character of the state in the training. I believe there might just be a high density of states in some energy range that biases the training process.
4. In the dynamics part, the impact of the new model is limited. There are no particular advances with respect to previous molecule-specific training for nonadiabatic dynamics. The model is not transferable and universal because it needs fine-tuning at a different level of theory, coming back to the problem of single-molecule training. It could proven a computational speed-up, but not a paradigm shift in ML application in NAMD. Additionally, I wonder why the work previously published in Nat. Comms. by Axelrod et al. is not cited, where an example of transferable ML architecture for NAMD is actually reported. (<https://www.nature.com/articles/s41467-022-30999-w>). Also more recent example of transferable NN for excited states should be mentioned (<https://arxiv.org/abs/2502.12870>).
5. The fulvene dynamics look very different from the reference in literature, where a reflection in the S_1 population is present, particularly with trajectory surface hopping (Phys. Chem. Chem. Phys., 2020,22, 15183-15196). This is due to a physical reason, with the molecule encountering two different conical intersections, one of them populating back the first excited states. This is the reason why this model has been chosen to test nonadiabatic methods. Already the reference method used in this manuscript misses this effect. The training with the model actually fully deletes the small reflection at around 15 fs, removing completely the physical behavior expected. I believe this is more due to interpolation reasons than capturing the complexity of the potential energy surface, but in general is an example where the prediction erases the correct physical behavior and potentially leads to misinterpretation. This could be solved using another formulation of TSH, for which nonadiabatic coupling or wavefunction overlap would need to be predicted by the model, but is not available in the architecture presented in this work.
6. More information about the computational cost of training would be necessary. Yes, the prediction is definitely cheaper than TD-DFTB, but working on a large dataset, what is the computational cost of the training with respect to calculating these excitations with the semi-empirical method?

Minor points:

- Is the label of Figure 4a correct? I think the authors show the ground state energies.
- The statement "While these methods are certainly more affordable for systems with hundreds and even thousands of atoms, the accuracy of their predictions can often be questionable" needs a reference.
- Some self-glorifying strong claims like "This marks the watershed moment that ML approach for excited-states presents a viable alternative to established QM methods." should be toned down, as they just represent the author's personal opinions.

(Remarks on code availability)

Reviewer #3

(Remarks to the Author)

This communication presents a novel foundation model / universal model for the excited state potential energy surfaces of organic molecules. The introduction motivates the model well in terms of deficit of the capabilities of existing computational methods for excited states in combination with high-throughput or dynamics-based studies of anything beyond very small molecular systems. The approach taken is to adapt a variant of the well-established ANI neural network potential, recently introduced by the current authors and other collaborators in [54] npj Computational Materials 11, 132, which introduces a state-index as an extra input to the neural network alongside the geometry and atomic numbers, resulting in a "MS-ANI" approach. The current work builds on the demonstrations in [54] by showing that a foundation model, OMNI-P2X, can be built with this architecture which can be general across a fairly wide range of organic molecules. Overall the model is demonstrated to be quite successful and I think the paper could be quite influential as it represents a first of its kind.

However, I feel there is room for improvement in the presentation and demonstrations before publication: in several sections the results themselves are very briefly described and it is hard to discern the level of success of the model.

Training data has been sourced from two existing databases, the PubChemQC database of TD-B3LYP/6-31+G* calculations, and the ANI-1ccx dataset of small molecules including off-equilibrium configurations, which has been provided as ground and excited-state DFT and TDDFT data for 10 excitations, and as ground-state-only CCSD(T), allowing the model to predict multiple levels of theory. Results show the S0 ground state has an MAE on energies of 2.63 kcal/mol which makes it comparable to other models trained on large databases of organic molecules, though I would prefer to see an explicit comparison to understand clearly how good the model is for the ground state.

The S1 state and higher-lying excitations are substantially worse, at ~5 kcal/mol, which is still useful, but it would be interesting to understand how correlated are the errors between S0 and the other states, ie should we expect errors on S0->Sx transition energies to exceed those of the contributing states, or will there be error cancellation to some degree? Scrutiny of Figs 4b and 4c might provide this information in an averaged sense, but they have different energy units so the comparison is not straightforward (generally, I find the broad selection of energy units (variously Ha, eV, and kcal/mol, sometimes in the same figure) rather troubling.

Figure 3 shows the distribution of sizes and excitation energies of the training data, indicating that the majority of the PubChemQC data is in the 20-40 atom range whereas that of the ANI-1ccx data is much smaller, peaking at 12 atoms. Figure 4 shows some test results: I note that while figure 4a purports to demonstrate close agreement between predicted and target total energy, the spread of energies is so wide (1000 Hartrees!) that presenting this as a y=x figure is highly redundant – any deviation from y=x visible at this scale would represent an enormous error, and the MAE of 5.78kcal/mol ~ 0.01 Hartree would be invisible on this scale. Perhaps a baseline subtraction of the atomic energies from each case would make this graph more useful?

Regarding the fact that 10 excited states are calculated: are this many states always going to give meaningful results in these systems with a 6-31+G* basis: in small molecules these might be higher-lying Rydberg states which are not well-represented in a relatively small basis such as 6-31+G*?

It is interesting that the Energy MAE on S1 is roughly double that of S0, then further higher-lying excitations have increasingly lower errors. It strikes me that perhaps the increasing Rydberg character of higher-lying states is somehow easier for the model to capture. The authors say “An outlier here is S0, which has the smallest energy MAE of 2.63 kcal/mol, which we attribute to the effect of the added ground-state CCSD(T) data” – I couldn't see how that could come about – why would the presence of data associated with a second level of theory predictable by the same model improve the accuracy of its predictions for the first?

Some minor points on figures: “One can note, that the highest accuracy is achieved for the S0 excitations to S1 and to higher-lying S8–S10 states.” Based on Figure 4d this does not appear to be true – do they mean the lowest accuracy? In the stick plot in Figure 5: the baseline of sticks does not align with zero on right hand axis – this needs correcting. Panel 8a is very separate from 8b,8c: they refer to different systems and unrelated calculations.

I would say that the Statement “OMNI-P2x is approaching the accuracy of time-dependent density functional theory” in the abstract and main text is potentially misleading, despite being carefully written. As Figures 5 and 6 show, while the model does significantly better than semi-empirical methods, at reproducing TDDFT spectra, as measured by the Spearman Correlation Coefficient, the reasons for this in terms of agreement of the underlying stick spectrum with TDDFT are often dubious. In many cases, where TDDFT has a few strong peaks, OMNI-P2X has a forest of intermediate peaks in a similar energy range, with no clear relationship between oscillator strengths. I would want to know what the Mean Relative Error on oscillator strengths was, as well as the MAE, because it seems like in many cases it might be quite large.

For a method to be able “approach the accuracy of TDDFT” I would expect individual excitations to be predicted to better than the 5kcal/mol quoted in Figure 4b as the MAE on S1 energies, and for the mean relative error on oscillator strengths not to be too extreme. This current level of energy error corresponds to 0.22eV or 100nm on a S0->S1 excitation around 1.5eV, which is not really approaching TDDFT accuracy.

The section on design and screening of azobenzene compounds provides a proof-of-principle for use in screening large datasets of candidate molecules. In this case azobenzene derivatives are screened to find 21 leads with strongly red-shifted absorption, at least some of which stood up to further scrutiny via calculations at a higher level of theory, ADC(2). I found it frustrating that this section ended without serious investigation of whether OMNI-P2X had really found strong leads: no comparison is made of the spectra from TD-DFT, from ADC(2) or from OMNI-P2X. I would want to see this investigated more deeply.

The final section demonstrates what could turn out to be a common use-case for foundation models for the excited state: active-learning-driven fine-tuning / transfer-learning is used to train a OMNI-P2X model at the CASSCF(6,6) level of theory for fulvene. The transfer-learning is shown to substantially accelerate model training, and both the TL and no-TL versions accurately reproduce dynamics of the reference CASSCF calculations. Very brief mention is made of calculations on photoisomerisation of azobenzene, but the results presented are rather cryptic: I was not able to intuit what point was being made about dihedral angles as obtained from the dynamics.

One natural question would be how the model responds to large molecular systems and relatively large numbers of states

required, and is there consistency of the predictions with respect to the number of states requested. In some testing of the version made available on figshare, I observed that when calling OMNI-PX2 three times, for 2, 5 and 10 states, for a model of a sunscreen candidate molecule with 95 atoms, in each case I got a very different set of excited states predicted (the S1->S0 transition was much lower in the 10 state case than the 2 state case). Is there anything that guarantees consistency with respect to the number of states requested. I note it is relatively stable for the provided example of benzene, though the oscillator strength varies a little.

Overall, the paper presents a novel methodology and opens up new avenues for investigation, in that it presents the first successful universal excited state model. However, I feel many of the tests and examples need to be refined further and presented more clearly before the community can properly judge the success of the model.

(Remarks on code availability)

I downloaded the shared version from figshare, verified that the benzene model reproduced the data provided, and verified the availability of the model's training data. I am not able to review the 25k lines of code associated with MLAtom itself, but the reproducibility was quite straightforward and MLatom worked out of the box in via torchani.

Reviewer #4

(Remarks to the Author)

(Remarks on code availability)

I sat with my supervisor and tested and discussed the code. I did not read the entirety of the codebase.

Version 1:

Reviewer comments:

Reviewer #1

(Remarks to the Author)

Thank you very much for the detailed reply to my comments. After the revision, I think the limitations are more openly communicated and indicated in the manuscript. As the manuscript is certainly a step forward towards generalizable prediction of excited state energies, I support the publication in Nature Communications.

Details:

Dimers: Thank you for the additional experiments. The fragment-correction scheme is very interesting and a valuable addition to the manuscript.

Level of theory encoding and excited states encoding: Thanks for the additional ablation study.

Out-of-equilibrium: Thanks for the very interesting additional tests, particularly Figure S10 and the fine-tuning study.

Figure 4: Thanks for the changes.

Figure 6: It is a very interesting observation that out-of-equilibrium errors are higher for lower excited states. Thanks for adding this information.

Delta-Learning: I understand the complexity of this topic, and I am looking forward to seeing a more detailed study of this. I agree that it might warrant a separate paper.

Transfer learning: Thank you for making this clearer and adding the new figure to indicate the impact of transfer learning.

(Remarks on code availability)

As shown in my first review of the manuscript, I used the code and found it to be well documented and easy to reproduce (at least the most basic tests).

Reviewer #2

(Remarks to the Author)

I thank the authors for addressing my concerns. I acknowledge the value of the new section on nuclear ensemble approach spectra, the addition of the nonadiabatic dynamics (NAMM) of azobenzene, and the discussion of some of the limitations of

the model. I believe that the revision improved the manuscript.

However, I believe that my chemistry-related comments have not been sufficiently scientifically addressed. Especially, if the model has already received high visibility as the authors report, it is even more important to ensure that it delivers physically correct results.

Additionally, I suggested some potential modifications that might make the impact of the model strong enough to recommend publication, like the prediction of the character of the excited states, nonadiabatic coupling, or spin-orbit coupling. However, none of these have been addressed, although in some cases, even acknowledged that architecture is inherently extensible for this. That was not only required to show a justification of the novelty of the model to be published in Nat. Comm., but also to avoid potential flaws in the chemical interpretation that can be obtained by using the model as it is.

I would like to justify my claims about the problems that the authors addressed and that persist in this revised version of the manuscripts.

1) The authors say:

“However, many (if not most) machine learned interatomic potentials (MLIPs) that are considered universal are limited to a certain set of elements, larger or smaller than OMNI-P2x.”

I agree with the authors. However, not clearly going beyond the state-of-the-art in this regard, it shows indeed the limited impact of the model in terms of offering a new paradigm for using MLIP for computational chemistry predictions.

2) My concerns about the correctness of the results used for training the model have not been sufficiently addressed or disproved.

The authors say: “The choice of reference method is obviously important and we fully acknowledge that TD-DFT/B3LYP has known limitations in describing certain excited-state properties, particularly charge-transfer and Rydberg states. Nevertheless, B3LYP remains widely used in photochemical studies and provides a practical balance between accuracy and computational feasibility.”

First of all, I disagree with the statement. Although it could reproduce good energies for some bright states, the danger of using B3LYP in excited states calculations over its benefits is very well established. That's why several range-separated hybrids, or functionals with higher percentages of Hartree-Fock exchange, are strongly recommended to be used for excited states calculations, and they do not add any relevant computational expense. However, the widely used approach in literature does not justify its correctness, especially because the performances of density functional approximations are highly case-specific. It would have been necessary to prove the accuracy of B3LYP with higher reference calculations or a systematic evaluation of the performance. This would have been very easy to show, for at least a pool of selected molecules, but it has not been done. Unfortunately, acknowledging the limitations of B3LYP, but saying that it is highly used, without showing a rigorous evaluation of its prediction, does not support the claim supported by scientific evidence and limits the trustworthiness of the accuracy of these calculations that are used to train the model and consequently its reliability to be used for unseen molecules.

However, I thank the author for addressing this problem in the discussion.

3) Additionally, the authors claim: “Moreover, to the best of our knowledge, TD-DFT is the only truly “black-box” electronic-structure approach currently capable of providing excited-state data for a broad range of molecular sizes and compositions across chemical space without requiring case-by-case adjustments to the calculation protocol.”

This is not true. ADC(2), available in several software (ORCA, TURBOMOLE, Q-CHEM) is black-box and more accurate. Its error is well-known, it does not require any case-by-case adjustment, and it does not depend on the mathematical form of the functionals chosen. It is yes, more expensive, but I believe affordable for the pool of molecules used in this work.

4) Regarding the UV/Vis spectra task, the conceptually inaccurate, strong limitation of being just a prediction of a collection of energies and oscillator strengths with poor correlation, with no actual information and correlation with states' character or wavefunction or orbitals involved, remains after revision.

The authors acknowledge this limitation and justify it in their response:

“While this limits interpretability in terms of electronic structure, it is consistent with how UV/Vis spectra are typically constructed from ab initio calculations, where oscillator strengths are reported for adiabatic states without explicit orbital labeling. The utility of OMNI-P2x is therefore in reproducing excitation spectra (energies and intensities) efficiently and at scale, rather than in providing mechanistic orbital analysis.”

Here, I again disagree with the authors. Even if there is no explicit orbital labeling, an ab initio calculation includes molecular orbital and excited configuration coefficients that allow for determining the character of the excited states. Additionally, well-established post-processing codes like Multiwfn or TheoDORE can efficiently analyze wavefunctions and provide descriptors to train a machine learning model for excited states. This is a fundamental of computational spectroscopy, where the calculations are needed to explain what is behind the shape of the UV/Vis spectra. If this information is missing, the scope of computational chemistry is strongly diminished. That's why I believe that a ground-breaking architecture should be able to properly reproduce the spectrum because it reproduces correct energy, oscillator strengths, and wavefunction information, not just the shape. If this model deliberately ignores this aspect, it not only reduces the impact of computational chemistry but also its reliability in terms of the physical interpretation of the results.

5) The authors write: "We note that, contrary to your comment, the $S_0 \rightarrow S_1$ excitation energies have consistently exhibited the lowest accuracy in both the original and revised models, so we are unsure where the claim of "the highest accuracy" originated. "

I just wanted to clarify this point. If the character and order of the excited states are not the target of the prediction, then, in cases reported in Figure 5a,b,c, the prediction is quite acceptable. The shape and the position of the maxima are ok for a qualitatively low-cost prediction, so it is not a problem with the prediction of the lowest states.

6) My comments about the NAMD example have been insufficiently addressed. I pointed out to the authors the reasons why this molecule was shown, implying a deeper physical analysis to justify the correctness of the simulation predicted. The trajectories are not analyzed to prove that the geometrical evolution of fulvene is well reproduced, passing through the correct conical intersections, and not only the time at which the hop should occur, or the energy gap. This is very important, especially because nonadiabatic couplings or wavefunction overlap are not predicted. I even suggested this point to the authors (since it is done in other implementations). I think this has not been properly addressed in the short paragraph in the discussion that was added.

Indeed, although the shape of the population resembles the CASSCF profile, there is no example of geometrical analysis, no check if the trajectory really passes through the correct conical intersection, or whether it's just the interpolation that correctly predicts the hops. The populations alone are not a metric of accuracy of a NAMD simulation, as highly sensitive to the simulation parameters, and they are not a measure of correct physical behaviour, as the decay from the excited states could be derived by a wrong deactivation channel. This is particularly important to show, especially because in the previous task on azobenzene, it was shown that out of the equilibrium, the prediction gets significantly worse. Same for the excited state character, there is no analysis to show that there is no problem in this example. Since in NAMD by definition geometries out of the ground state equilibrium are explored, and the excited states' character will change, this check is fundamental, in my opinion, to show the reliability of the model. It is missing the proof of the correct physical behaviour beyond just interpolation. Additionally, in figure 9, I believe that the y-axis should be the S_1 population.

7) I appreciate the addition of a second experiment regarding the NAMD. However, the analysis is rather limited, and again, it is not shown, either in the SI, that the propagation that is delivered is correct. More importantly, also considering this example, I once again acknowledge the computational speed-up of the new active learning procedure for NAMD. But this is an improvement on how active learning is designed, but not a paradigm shift for how ML potentials are used in NAMD, since there is still a need for fine-tuning, and the model is not transferable for this task. Additionally, the lack of couplings, available in other architectures, does not push towards a physical improvement in the way MLIPs are used for NAMD.

In conclusion, while I agreed with the other reviewers about the advancement in terms of architecture, I also saw similar problems in the results shown. In my first revision, although I recognized that the model works, I underlined some important, potentially wrong interpretations that could be derived from the predictions of the model in computational chemistry applications. These have not been disproven, and the limitations have not been addressed, but remain after revision. Since I find them highly relevant, I still cannot recommend publication in Nature Communications, although I believe this version of the manuscript is improved.

(Remarks on code availability)

Reviewer #3

(Remarks to the Author)

The authors have given responses to each of my comments, and made substantial changes to the manuscript. Many of the concerns I and other referees had are addressed in a fully satisfactory manner, but some of the most important are not, as detailed below. Overall I would say the authors have demonstrated that the model shows some promise for applications in theoretical spectroscopy and non-adiabatic dynamics, and represents an interesting methodological step forward, and the ML model shows real promise.

However, all reviewers have expressed a range of different concerns about its accuracy and the overselling of the capabilities of the current model. These concerns add up to an uncertainty that the model exhibits the level of accuracy that would be necessary for an out-of-the-box tool of genuine usefulness to practitioners not expert in analysis of the accuracy of ML models. Given that it has been presented as a ready-to-use general-purpose tool I am therefore worried that publication in Nature Comms would be giving a seal of approval on the accuracy of the model that the current results do not support.

This is particularly important given the strong statements about the accuracy, some of which the authors have not significantly toned down despite referees concerns, and which it is very important to avoid being exaggerated. Examples include "making ML approaches for excited-states a viable alternative to established QM methods" and "The model enables out-of-the-box predictions of electronic spectra of organic molecules, with accuracy approaching TD-DFT/B3LYP". Uncritical use of the model as a substitute for TD-DFT calculations would be extremely risky, given the current results. Therefore I would not support publication of the manuscript in its current form.

Comment 1

Note that there is continued mixing of units even when 2 of the referees asked them not to do this as it is confusing for the reader. This unit mixing creates the appearance of deliberately obscuring the substantially worse performance of OMNI-P2x with respect to ground-state universal models, which if I am reading it correctly, appears to be a very substantially higher error (albeit with less training data). That is not consistent with statement that “they demonstrate that the ground-state error of OMNI-P2x exhibits decent performance, even with respect to the current state-of-the-art models”

Comment 2

I find the decision to replace the model entirely with an ensemble with reduced errors at this stage somewhat surprising – has this substitution been made consistently across all the data in the paper, or just applied selectively to places where the referee concerns were most serious?

The statement “The excitation energy errors are similar in magnitude to the target state energy errors and follow the same trend” doesn’t really answer the original question of “should we expect errors on $S_0 \rightarrow S_x$ transition energies to exceed those of the contributing state?”.

Comment 3

Figure 4a has been improved and is now more useful in determining the spread of errors.

Comments 4 and 5

Appropriate cautionary language has been inserted to address the problem of predicting as many as 10 excited states for small molecules, and the fact that increasing success may be due to increasing Rydberg character.

Comment 6-8

Minor presentational issues raised here have been acted on appropriately.

Comment 9

The main point of this comment was that the statement that the model approaches the accuracy of TDDFT is not justified by the data. The reason to use a ML surrogate model is to bypass the computational expense of a given high or moderate level of theory such as TDDFT or QC methods by replacing it with a model that gives an error compared its ground truth that is substantially lower than the error of the ground truth itself: otherwise one is simply compounding the ground truth error. The authors response is simply that the model makes a further error (on top of the TDDFT error) that is comparable to typical errors of TDDFT. I would say this is not acceptable in an ML surrogate model, and significantly undermines the possible uses of the model, and I would want to see the statement removed or revised substantially.

Comment 10

This section is more satisfactory now that the SPC spectra for the top 3 candidates is shown. These spectra show there is not particularly good agreement between OMNI-P2x spectra and TDDFT, but the presence of low-energy redshifted absorption in both does at least mean that the screening task of finding such molecules has been successfully achieved.

Comment 11

The new Section S4 of the ESI is very cryptic and does not add to the discussion. Taken at face value S12a simply compares LZBL with FSSH and shows they agree for the same PES, which is no great surprise and gives no information about the quality of the PES itself. Furthermore, Referee 2’s concerns about the accuracy of the dynamics do not appear to have been fully addressed in the rebuttal. However, the demonstrations now included in Figure 10 are fairly convincing evidence that the TL based on OMNI-P2x is effective in significantly reducing the amount of training data required to converge the model predictions with respect to training set size in such calculations.

Comment 12

The fact that the revised model issues warnings when the predictions are highly uncertain (as assessed by committee standard deviation predictions, presumably) is a significant improvement on the original approach.

(Remarks on code availability)

Reviewer #4

(Remarks to the Author)

(Remarks on code availability)

Version 2:

Reviewer comments:

Reviewer #2

(Remarks to the Author)

I thank the authors for addressing my comments and improving the manuscript, especially thanks to the new analysis reported.

In particular, the atom-wise energy decomposition and the geometrical analysis of fulvene dynamics make the interpretation of the results more reliable and robust. However, for the atom-wise energy composition analysis, a general metric to evaluate the accuracy would be more suitable than potentially cherry-picked examples. The comparison between B3LYP and ADC(2) is also important, showing that, although correlated, the performances of TD-DFT are systematically worse than ADC(2).

I think the manuscript is highly improved through the two revision rounds, and I have no additional comments for the authors that prevent me from recommending publication.

(Remarks on code availability)

Reviewer #3

(Remarks to the Author)

I do not have any new comments to add in relation to the latest revision.

(Remarks on code availability)

Reviewer #4

(Remarks to the Author)

(Remarks on code availability)

Reviewed with supervisor.

Responses to the comments of the reviewers

Reviewer comments are in *italics*, and our responses are in **regular, blue font**. Our changes to the manuscript are marked in **red font**.

Reviewer #1

The manuscript introduces OMNI-P2x as a universal neural network potential for molecular excited and ground state energy prediction. The model is inspired by the multi-state ANI model, based on own prior work by the authors [54], and as a novelty in this manuscript, adds all-in-one learning to simultaneously learn from multiple datasets with different levels of theory.

The manuscript is a step forward towards generalizable prediction of excited states energies, but there are a few questions that should be discussed before publication, in order to not only emphasize the strengths but also the limitations of this work.

We thank you for your positive feedback. In the following revision, we have put great effort into increasing the clarity of the presented results, showing applications that better showcase the power of the OMNI-P2x architecture, especially in NAMD simulations, as well as clearly stating the limitations of the presented work. Furthermore, further computational effort allowed us to double the training set size of the ML model and replace the NN potential with an ensemble of three potentials. This allows not only for more accurate and stable predictions of ground and excited state energies, but also to quantify the uncertainty of the NN predictions.

For the energy part of the predictions, an ensemble of three NN potentials is used, with their predictions averaged to obtain the final predicted energy for all electronic states.

We have used the entirety of the PubChemQC dataset, which meets our filtering criteria (as detailed in the Methods section), yielding 3.1 million molecules with 11 electronic states each.

Additionally, OMNI-P2x provides an uncertainty quantification (UQ) of all predicted energies, using the standard deviations between quantities predicted by each of the three NN potentials as the error bar. A warning

is issued to the user if the UQ of any excited states exceeds a safe threshold, which is determined as the median plus three median absolute deviations of the UQs of the test set, for each electronic state.

Reviewer 1, Comment 1

Page 6: "The predictions are then made atom-wise and summed up to yield the total energy of the molecule."  Does that mean that also the excited states energies are predicted by a sum of atom contributions? How are the labels for the excited states pre-processed (relative to the total energy of the ground state or as absolute energies), and how are they normalized?

Yes, excited state energies are predicted in the same way as ground-state energies, which is a sum of atom-wise contributions. The energy labels of all states are pre-processed together, by means of computing one set of self-atomic energies in the ANI model that is used for all states and then subtracted. This ensures that the energy gaps between electronic states are conserved in the raw ANI energies, which would not be the case if self-atomic energies were computed for each state separately, as tested in our previous work developing the multi-state ANI model.

Reviewer 1, Comment 2

A simple test with your code and a duplicated and shifted benzene molecule shows that the excitation energies double when duplicating (and shifting) the input molecular structure. Of course, the model was not trained on such an input system, but the inductive bias given by the summation of single-atom contributions is not suitable for the prediction of excitation energies as non-extensive quantities (in contrast to ground state energies):

```
import mlatom as ml
omni = ml.models.methods(method='omni-p2x')
mol = ml.data.molecule()
mol.read_from_xyz_file("Benzene.xyz")
omni.predict(molecule=mol, nstates=5)
mol.excitation_energies
0.18915822, 0.20142478, 0.2106019, 0.21771565
mol2 = ml.data.molecule()
mol2.read_from_xyz_file("Benzene2.xyz") # two benzenes, the second is shifted by 10A
in +z direction.
omni.predict(molecule=mol2, nstates=5)
mol2.excitation_energies:
0.37831637, 0.40284953, 0.42120379, 0.4354313 # roughly twice the excitation
energies of the single benzene molecule.
```

We thank you for pointing out this potential problem and providing a code snippet allowing for its reproduction. OMNI-P2x uses the ANI framework for predicting electronic state energies, which generates atom-wise atomic environment vectors (AEVs) for each atom. These AEVs include information about neighboring atoms up to some cutoff (set to 4.0 Å in this work). If parts of a system are completely separated by a distance larger than this cutoff, like in the benzene dimer example, such pathological behavior can be expected, as the descriptors of each monomer have no information about the rest of the system. To simulate this behavior, we have computed the $S_0 \rightarrow S_1$ vertical excitation energy for a benzene dimer as a function of the monomer distance, R , which is presented below.

We can note that while the ML-model initially gives predictions that are reasonably close to the DFT results (considering the fact that dimeric systems were not explicitly included in the training set), for distances between 3 and 4 Å the excitation energy begins to rise, as parts of the system move out of the cutoff radius of the rest, eventually reaching about double the excitation energy of TD-DFT. From this, we conclude that while OMNI-P2x cannot correctly describe excitation energies for systems with large spatial separation due to the intrinsic limitations of the AEV descriptor, it can implicitly learn the correct, intensive character of the excitation energy, within the descriptor cutoff.

In the limit of infinite separation of chromophores, the OMNI-P2x's solution will indeed collapse into the single excitations on each of the chromophores. This is unphysical, but it is also an issue with some of the first-principles calculations, i.e., the calculations of a multi-chromophore system in periodic boundary conditions, also require deciding which chromophore to treat as the excited one, while others are treated as non-excited chromophores.

Importantly, the NN architecture itself *can* learn the correct behavior for separated systems, provided suitable training data. To demonstrate this, we fine-tuned OMNI-P2x on a set of 600 benzene dimer conformations sampled via metadynamics, with the results shown below. The agreement between the ML prediction and reference is excellent.

Despite this capability, we acknowledge that the absence of such systems in the original training data, as well as the inability of the model to infer correct long-range excitation behavior from monomer data alone, results in OMNI-P2x producing unphysical predictions for widely separated chromophores.

Hence, to correct the long-range unphysical behavior, we propose a mitigation strategy, where we automatically decide which chromophore to treat as excited, while the others are treated in their ground state. Before predicting for a given system, a check is run to detect non-covalently bound fragments of the system. Then, predictions are made separately for each fragment and the entire system. If the excitation energy of the total system is greater than the minimum excitation energy of the subsystems, that means that OMNI-P2x most likely incorrectly treated the system as multiply excited, with an excitation localized on each chromophore. Hence, calculations are performed separately for each sub-system and sorted to yield the final spectrum. If, however, the excitation energy of the total system is smaller than the minimal energy for each sub-system, we identify it as an excimer and keep this result. A visual representation of the fragment-correction algorithm is given below:

We compare the results obtained with the fragment-corrected approach to pristine OMNI-P2x and TD-DFT below.

The agreement between TD-DFT and the fragment-corrected scheme becomes excellent for distances above the local descriptor cutoff. It should also be noted that DFT is not the most suitable level of theory for describing systems with large spatial separation, which is also evidenced here by an unphysical bump in the excitation energy between 8 and 9 Å.

The revised manuscript now reads, with the above discussion recounted in section S1 of the ESI:

A fragment-correction scheme is applied to non-covalently bound structures, to recover the correct, intensive character of excitation energies, for molecular fragments separated by distances greater than the descriptor cutoff radius, with details provided in Section S1 of the ESI.

Reviewer 1, Comment 3

Figure 2: How exactly is the state and the level of theory encoded? Figure 2 suggests that the state is an integer number appended to the molecular geometry vector, while the level of theory is represented as multiple one-hot encodings of length two (which seems not intuitive). Why is the state not also represented as a one-hot encoding? Do you have empirical evidence that this is the most informative / data-efficient representation?

Your interpretation of Figure 2 is correct, the state ordering number is indeed an

integer that is concatenated with the atomic environment vector (AEV), while the level of theory is a one-hot encoded vector. We use non-negative integers to encode the state, as the adiabatic state energies are in ascending order, the same as the descriptor used. On the other hand, there is no clear ordering between the energies predicted at different levels of theory; hence, we choose a one-hot encoding. To have a direct comparison, we have trained two separate NN models on the fulvene dataset presented in this manuscript: one using integer encoding for states, and the other using one-hot encoding. The integer-encoded model outperformed the one-hot-encoded model, achieving a MAE of 0.0279 and 0.02590 for S0 and S1 energies, while the one-hot-encoded model had MAEs of 0.0330 eV and 0.0326 eV, respectively. We take this benchmark, along with the physically motivated ordering of the descriptor, as evidence to use it in our foundational model.

The manuscript now reads:

The electronic state is encoded as a non-negative integer, consistent with the natural ordering of adiabatic state energies, which we find to be both physically motivated and empirically more accurate than one-hot encoding.

Reviewer 1, Comment 4

Page 7: "PubChemQC project's dataset consists of unique, optimized structures of molecules at the TD-B3LYP/6-31+G level of theory, with the energies and oscillator strengths of the first 10 excited states calculated using TD-DFT."  How does the model without transfer learning perform on excited states of equilibrium and non-equilibrium structures? Does the model without transfer learning just apply the learned ground state PES also to the excited states, so does it overestimate the correlation between ground and excited states, i.e. parallel ground- and excited states surfaces? Or does it somehow learn to predict excited states PES (at least close to equilibrium structures), even though it was only trained on excited states energies at equilibrium structures?*

Thank you for yet another interesting question. To check this, we have performed two relaxed potential energy surface scans, with results presented below. Panel a) shows the scan along C=O bond stretching in cyclohexanone, while panel b) shows a torsional scan around the C-N=N-C dihedral angle in azobenzene. The geometries were optimized using DFT/B3LYP(6-31+G*), and results obtained using TD-DFT/B3LYP and OMNI-P2x (DFT-level predictions) are compared for the first three electronic states. For the cyclohexanone example, as expected, in the vicinity of the geometric minimum (1.2 Å), the

model is very accurate. This accuracy begins to lower as we move outside of the minimum, although for the stretched C=O bond the accuracy is still rather reasonable, without any catastrophic extrapolation failures. Shortening of the C=O bond however leads to much larger errors, as such high-energy structures are far from the data contained in the training set. Taking a look at the more complicated example of azobenzene torsion, around the two minima (0 and 180 degrees), OMNI-P2x achieves near perfect accuracy for the ground state, and an error of about 0.25 eV for the S1 and S2 states. This accuracy is retained up to around 25 degrees outside of the minimum, where it starts to rapidly decay due to the change in character of S0 and S1 associated with twisting of the N=N double bond, which cannot be captured by OMNI-P2x, due to the lack of corresponding structures in the training set. From this test, we conclude that despite training on just equilibrium structures, OMNI-P2x can extrapolate the shape of the excited-state PES's at points close to the geometric minimum; however, when this geometric change is associated with a substantial change in character of the excited state (such as in the azobenzene case), this extrapolation fails. The ground-state and excited-state PES are not parallel, although, in line with your suggestion, their correlation is certainly overestimated when moving to out-of-equilibrium structures.

The main text has been modified to read:

To further probe this behavior, we performed relaxed PES scans of cyclohexanone (C=O stretch) and azobenzene (C–N=N–C torsion), shown in Figure S10. These scans confirm that OMNI-P2x is accurate near equilibrium and along moderate distortions, but errors grow for strongly compressed geometries or when torsion induces a change in excited-state character. Together, these results indicate that OMNI-P2x can extrapolate PES shapes close to equilibrium, but its reliability decreases as the molecular distortion drives the system beyond the domain of the training data.

At the same time, we show that the capabilities of the model can be easily extended to non-equilibrium structures by fine-tuning, even in the ultra-low data regime. Another example on which this is demonstrated, aside from NAMD, is by simulating nuclear ensemble approach spectra, in a new section “Nuclear ensemble approach spectra”. By transfer learning with just 25 datapoints, the model can perfectly recover the dynamically averaged spectrum of pyrazine, and with just 50 points can produce accurate results for the much more complicated example of 9-DCMA. Noteworthy, this example contains 31 electronic states, while OMNI-P2x was pre-trained on only 11. Despite that, the accuracy of the higher excited states is acceptable (with near-perfect reproduction of the lower energy region of the spectrum), having correct placement of the peaks and their relative intensity. This hints that the model can also extrapolate in terms of the number of electronic states requested. Furthermore, these results show unprecedented data-efficiency, as previous approaches required at least 100 to 200 points to produce acceptable results. The main text now contains a new section, as follows:

A more accurate method for predicting UV-VIS absorption spectra is the Nuclear Ensemble Approach (NEA). In this technique, vertical excitation energies and oscillator strengths are computed for an ensemble of hundreds or thousands of molecular geometries. The final spectrum is constructed by summing Lorentzian or Gaussian functions centered at each geometry’s excitation energy, weighted by the corresponding oscillator strength. Compared to single-point convolution, this approach better captures vibronic effects. However, NEA simulations are computationally demanding, as they require quantum mechanical (QM) calculations for all structures in the ensemble. To address this cost, machine learning (ML) models offer a natural path for acceleration.[78, 79] Since OMNI-P2x is initially trained only on equilibrium structures, it is not directly applicable to NEA simulations. Nevertheless, fine-tuning enables it to achieve remarkable data efficiency and simulation speed in this context.

To demonstrate this, we consider two test cases: pyrazine (with 4 excited states) and 9-Dicyanomethylene-Acridine (9-DCMA, with 30 excited states), the latter of which was previously used to benchmark ML-NEA spectra with kernel ridge regression models.⁷¹ The NEA spectra simulated using reference TD-DFT data, as well as those predicted by the fine-tuned OMNI-P2x model—trained on just 25 data points for pyrazine and 50 for 9-DCMA—are shown in Figure 7. For pyrazine (panel a), the agreement

is excellent across the entire spectrum. In the more complex case of 9-DCMA, the model accurately captures the intensity and position of the lower-lying excitations. Deviations start emerging below 250 nm, especially in predicted intensities, while the position and relative intensities of the peaks remains rather accurate. We attribute this to the model extrapolating in a regime beyond the number of excited states included in training dataset: OMNI-P2x was pre-trained on 10 excited states, while 30 are considered in this example. Such simulations can be done on commodity hardware (Jupyter notebook using 4 CPUs), in under a minute for pyrazine, and about 40 minutes for 9-DCMA, excluding the labeling time of QM methods.

Reviewer 1, Comment 5

Figure 4a shows total energies, but it is unclear if those are ground-state or excited-state total energies.

Furthermore, calculating the r^2 score on non-normalized total energies is misleading. Even just applying the atom-specific normalization to total energies without learning anything will lead to very high r^2 scores. The r^2 score should be calculated on the predictions on the normalized scale, not on the unnormalized scale.

Figure 4 shows three different units. It would be better to discuss everything in one unit (e.g. eV), and then additionally convert key quantities to kcal/mol. But showing energies in Hartree in panel (a), MAEs in kcal/mol in (a) and (b), and then MAEs in eV in panel (c) is very confusing. Additionally, the difference between (b) and (c) does not become fully clear from the caption or text.

Excitation energy predictions (relative to the ground state, separated by state) should be plotted in scatter plots, and r^2 values should be calculated. Figure 3b indicates a standard deviation of ~ 1 eV for excitation energies in the dataset, but I assume that the std is smaller if this is separated by state. Figure 4c indicates MAEs of ~ 0.2 eV, which is significant compared to the variance of the labels. Therefore, I assume that the r^2 scores are not 0.9999, as calculated for the non-normalized ground state energies. A more open discussion about the actual accuracy (also in terms of r^2 scores) is absolutely necessary to judge the reliability of the results and the impact of this study.

We agree with the suggestions concerning the clarity of the results presented in Figure 4. In line with this comment, we have made the following changes to Figure 4:

- 1) Figure 4a has been changed to compare normalized energies (with self-atomic energies removed), with the R^2 calculated.
- 2) All units have been converted to kcal/mol.
- 3) Scatter plots of predicted vs true excitation energies for each state have been presented in Figure S4 of the ESI, with calculated R^2 scores and MAE. In all cases the MAE of the predicted excitation energies is smaller than the reported lower bound of the accuracy of TD-DFT ($0.2 \text{ eV} \approx 4.6 \text{ kcal/mol}$), making the error related to fitting the ML model lower than the intrinsic accuracy of the method.

The main text has been modified as follows:

We evaluate the performance of the model using a test set of 300k unseen molecules from the PubChemQC dataset for the excitation energies, as well as a test set of almost 49k ground-state conformations from the ANI dataset. The MAE in energy predictions for the first 11 electronic states with respect to the TD-DFT data shows that the quality

of the predictions increases when going to higher excited states, reaching a minimal value for S7 (Figure 4b and c). An outlier here is S0, which has the smallest energy MAE of 1.6 kcal/mol, which we attribute to the improved data-efficiency due to the application of the all-in-one learning to the combined set with the ground-state DFT (from PubChem) and the CCSD(T) (from ANI-1ccx) data, and an overall smoother and lower-variance distribution of ground-state energies that makes them easier for the model to learn. Furthermore, the 8 MAEs for both the predicted total and excitation energies are below 0.2 eV (4.6 kcal/mol), which lies below the lower bound of the reported accuracy for the reference TD-DFT method, indicating that the fitting error of the ML model is smaller than the intrinsic error of the underlying quantum chemical approach. Correlation plots showing the predicted vs true excitation energies for all transitions are available as Figure S4 of the ESI. In all cases, the R^2 is above 0.91. For the ground-state energies in the test ANI-1ccx subset, OMNI-P2x has a higher MAE of 4.14 kcal/mol, but still excellent $R^2 = 0.9957$ for the normalized energies (0.9999 for unnormalized).

Reviewer 1, Comment 6

Figure 6a/b fully focuses on equilibrium structures, which makes sense given the training data of the OMNI-P2x model, but which is not a fully fair comparison. Therefore, this should be explicitly mentioned in the caption and in the text/discussion. If possible, non-equilibrium structures of the same molecules should be sampled at a given temperature to see how fast the OMNI-P2x accuracy decays when going away from equilibrium geometries.

We revise the manuscript by changing the caption of Figure 6 to explicitly mention that the evaluation was done on equilibrium structures only:

The evaluation of the quality of the UV/Vis absorption spectra predicted by OMNI-P2x model on a test set of 50k molecules, at their ground-state equilibrium structures.

Sampling a representative number of out-of-equilibrium conformations that include finite-temperature effects for 50k molecules would require on the order of millions of additional reference calculations, which is unfortunately not computationally feasible within the scope of this work. However, to address this important point, we extend the manuscript with two complementary investigations. Firstly, the newly added “Nuclear ensemble approach spectra” section shows a workflow where OMNI-P2x can be used to predict spectra with explicit inclusion of finite-temperature effects, at a fraction of the cost, which has been discussed in detail in response to your Comment 4. Furthermore, the performance of OMNI-P2x when moving away from equilibrium structures has also been quantified by means of PES scans, also described in the

response to Comment 4.

Secondly, to directly assess the robustness of the base model when moving away from equilibrium geometries, we sampled 200 thermally distorted conformations of the molecules presented in Figure 5 and evaluated excitation energy predictions. The violin plots below (available as Figures S5–S8) show that while errors for higher excited-states remain close to equilibrium-structure errors, lower-lying states (S1–S3), are much more sensitive to thermal fluctuations, and thus, their errors can be much higher, with an overall 2 to 4 times lower accuracy of the prediction.

Those results are referenced in the manuscript as:

For the test molecules shown in Figure 5, we tested the effect of finite-temperature distortions on the accuracy of OMNI-P2x and found that higher excited states remain close to equilibrium benchmarks, while the accuracy of the lowest excitations (S_1 – S_3) may decrease more sharply. Violin plots showing the error distribution on non-equilibrium structures are available in Figures S6–S9 of the ESI.

And in the Methods section:

The thermally distorted structures of molecules presented in Figures S6–

S9 were sampled from a 2-ps-long ground-state Born–Oppenheimer molecular dynamics trajectory, with the first 1 ps discarded for thermalization, performed at the GFN2-xTB level of theory.

Reviewer 1, Comment 7

Figure 6b suggests that ODM2/CIS and TD-DFTB are not on the Pareto front of speed and accuracy. However, it would be interesting to see how a delta learning approach that combines ODM2/CIS or TD-DFTB with OMNI-P2x would perform. The cost is roughly the same as the semi-empirical methods (because OMNI-P2x inference time is negligible), but the overall accuracy might be above the accuracy of OMNI-P2x.

Indeed, this is an interesting question. At the moment of preparing this manuscript, we have been working on delta-learning models for excited states for a few months already. We observe that in certain cases delta-learning for excited states can outperform pure ML, but, unfortunately, it requires more careful considerations than delta-learning applied to the ground state problems, for which one of us has extensive expertise. In general, delta-learning for excited states brings about many additional questions and design choices, and requires a dedicated, in-depth study, which we are going to report separately.

Reviewer 1, Comment 8

All lines in Panel 8a seem more or less identical. It is not clear if the TL approach is statistically significantly better than the No-TL approach. A zoom into a particular part of the plot and the addition of uncertainty intervals is necessary to support the claims. Furthermore, the accuracy of TL and No-TP approaches in the population-time-curve should be quantified using an appropriate metric and plotted as a function of the active learning dataset size to obtain more insight into the advantages of TL here. The examples from 8c/d are not enough.

We realize that our original description and figures were not clear enough. We re-worked the fulvene example, running 10,000 trajectories with both ML models (TL and no-TL). The populations obtained using both the models, and reference CASSCF results have been plotted in Figure 9, along with 95% confidence intervals shown with shaded areas. An inset zooming in on the critical region between 10 and 20 fs has been added to the plot. As can be seen there, the populations obtained using transfer learning and direct ML models exhibit statistically significant differences in the vicinity of the conical intersection, and it is the TL model that stays within the 95% confidence uncertainty interval of the reference population, while the model trained in an AL loop starting from scratch produces results that differ from the reference.

The revised manuscript related to the fulvene dynamics now reads:

By employing transfer learning from a universal potential, we are able to slash the convergence time of fulvene from over three days, to about 24 hours, with a ca. three-fold increase in computational efficiency. Additionally, the final employed training set contains only 3550 conformations, which is about twice less as was necessary to achieve convergence in a reference direct active learning procedure, as can be seen in Figure 9. Moreover, inspecting the time evolution of the electronic state populations, we can see a statistically significant difference in the populations predicted by the fine-tuned and the previous MS-ANI models. Despite a smaller training set, the TL version of OMNI-P2x better captures the photophysical behavior of the system. Additionally, a comparison of the dynamics predicted using different ML models and different LZBL propagation settings with reference fewest switches surface hopping dynamics is available in Section S4 of the ESI.

To increase the clarity of the presented results, numerical experiments simulating the photoisomerisation reaction of azobenzene have been moved to

a separate Figure 10. We trained models using fractions of the full active learning dataset from ref [57] (consisting of ca. 35,000 points), ranging from 0.1% to 100%. For each fraction, a model was trained either using transfer learning (TL), or from scratch, and used to propagate NAMD starting from trans isomer of azobenzene. We then quantify the accuracy of the predicted dynamics, by calculating the photoisomerisation quantum yields, as well as the stability of the dynamics (measured as the fraction of the trajectories that remain undissociated until the end of the NAMD trajectory). The obtained results show that the transfer learning model can propagate stable dynamics even with as little as 0.1% of the training points, while the model trained from scratch requires at least 10% of the training set to achieve consistently stable results.

Furthermore, looking at the quantum yields of the photoreaction predicted by the two types of ML models, all of the TL models trained on a fraction greater than 2% of the training set predict quantum yields that are within a 95% confidence interval of the reference method (with the respective error bar), while the results obtained using ML models trained from scratch can deviate from the reference method, depending on the fraction of points use, indicating less stable results.

The revised manuscript reads:

To assess the improvement in data-efficiency that comes from using a pre-trained universal model, we compare the performance of the OMNI-P2x fine-tuned and MS-ANI trained from scratch on fractions of the full training set required to converge an AL procedure (35071 points, as reported in Ref. 57). Looking at the stability of the dynamics, calculated as the percentage of trajectories finishing without dissociation, shown in Figure 10a), the fine-tuned OMNI-P2x model can propagate stable dynamics given just 0.1% of the dataset, while dynamics propagated using the pure learning approach become consistently stable above 10% of the

data. Furthermore, Figure 10b presents the predicted quantum yield of trans to cis photoisomerization, as a function of the training set size. For the fine-tuned OMNI-P2x potentials, all models trained on above 2% of the data predict QYs within the reference range, up to a 95% confidence interval error bar. On the other hand, the performance of the models trained from scratch is much less consistent, with outliers predicting incorrect quantum yields even at high fractions of the training set.

Reviewer #2

In their manuscript, Martyka et al. report the development of two artificial neural networks that, combined, predict approximate TD-DFT excitation energies and oscillator strengths of molecules. This is obtained by combining existing and established architectures by including data at CCSD(T)/CBS and TD-DFT B3LYP/6-31+G levels of theory. The authors test their model on three applications, namely the convolution of absorption spectra based on predicted energies and oscillator strengths, the rapid screening of azobenzene derivatives to discover new candidates with desired properties, and the use of the model as a starting point to further fine-tune it with active learning to compute energies and gradients for nonadiabatic dynamics. While the manuscript introduces a potentially interesting architecture that contributes to increasing the potential of and reducing the limitations of artificial intelligence in the field of computational photochemistry, this work appears to have some theoretical limits and rather limited impact, breakthrough, and interest to justify publication in Nature Communications.*

Additionally, the authors claim the universality of the model, which would mislead the wide readership of this journal that might interpret the model as actually universal and flawless, which, in my opinion, it is far from.

We respect your opinion and constructive comments below, but with politely disagree with the claims of ‘limited impact, breakthrough’ – our models are already used in many contexts, not just in our groups, but have also attracted interest from other researchers. Reviewer 3 evaluated our work as follows: “Overall the model is demonstrated to be quite successful and I think the paper could be quite influential as it represents a first of its kind”. Our models are certainly not ‘flawless’, and we have emphasized that in the revised version of the manuscript. It should also be stressed that hardly any novel tool aiming to provide breakthrough functionality is free from flaws and limitations. Those limitations will likely be gradually removed by future work that aims to improve the current cutting-edge functionality.

The term ‘universal potential’ is commonly applied in literature in the context of potentials pre-trained on big data sets and generalizing to unseen compounds and it is general knowledge that they are not expected to be flawless. In this respect our models are universal as they generalize to unseen compounds, albeit not flawlessly.

Reviewer 2, Comment 1

There are three reasons why I disagree with the universality and the high impact of the model:

- The model seems definitely transferable (with debatable prediction accuracy) among the chemical space similar to the one used for the training. However, this is limited to a certain type of molecule (only containing H, C, N, O, F, S, and Cl atoms), leaving out large categories of important photoactive molecules. For example, no transition metal complexes are considered, whose inclusion would make the model more universal.

We agree that inclusion of more elements in the training set would make the model more universal. However, many (if not most) machine learned interatomic potentials (MLIPs) that are considered universal are limited to a certain set of elements, larger or smaller than OMNI-P2x. Examples include ANI-1ccx (4 elements), ANI-2 (7 elements), MACE-OFF (10 elements), and AIMNet2 (14 elements). Furthermore, the set of elements included in OMNI-P2x (H, C, N, O, F, S, Cl) covers the vast majority of organic chemistry, encompassing most molecules of practical interest. We are aware that certain experts only consider models universal if they cover most of the periodic table, but it remains a subjective opinion without consensus in the community as no terminology rules are officially adopted by any of the regulation bodies (such as IUPAC). We and many other groups consider potentials universal if they generalize to sufficiently large space of chemical compounds to which OMNI-P2x certainly belongs.

While excluding transition metals from the training set precludes the usage of the model for an important class of compounds, we would like to point out that most of ground-state MLIPs (all of the listed above) support only non-metallic elements. Hence, transition metals complexes remain challenging even for ground-state ML. This is due to the fact that these systems present fundamental challenges that go beyond adding new elements. Transition metals often exhibit multiple oxidation and spin states, strong multireference character, and significant relativistic/spin-orbit effects, making their electronic structure much more complicated. For these reasons, universal modeling of transition-metal

complex photochemistry may require dedicated datasets, ML architectures and descriptors, that are beyond the scope of the present work.

Reviewer 2, Comment 2

- There is no information about the character of the states, triplet states energies, nonadiabatic and spin-orbit coupling are not predicted, and there is no information about how to deal with ionic molecules. The model is able to predict vertical singlet excitations with a feature for the state order, but there is still no information about the orbitals involved and the actual character of the excited states. The universality is claimed to be in terms of how many different molecules can be predicted, but not in terms of photophysical properties and accuracy.

We acknowledge the limitations of the model. The current version of OMNI-P2x supports only molecules within the singlet manifold of states, and does not predict nonadiabatic, nor spin-orbit couplings. As almost all universal or non-universal MLIPs, it does not provide information about the molecular orbitals of a given molecule. We added a paragraph clearly stating the limitations of the model in the “Discussion” section of the manuscript:

While OMNI-P2x is the first universal excited-state neural network potential, it naturally has scope for further enhancements. Expanding the training set could enable support for a wider range of element types and out-of-equilibrium structures. Future extensions of the framework will aim to predict not only for neutral molecules in the singlet manifold, but also for states of higher multiplicity and charged species. Although the present focus on singlet states already enables accurate and chemically diverse predictions for many photophysical applications, the architecture is inherently extensible, and these additional capabilities will be the subject of future work.

Reviewer 2, Comment 3

- The training on the TD-DFT data makes for sure possible to predict energies and oscillator strength, but this is still training on wrong results in most cases, as the functional chosen is well known to have very poor reliability for the description of fundamental features of excited states. So a universal model should predict good and physically meaningful results, and not have as a goal to perform better than poorly reliable semi-empirical approaches.

The choice of reference method is obviously important and we fully acknowledge that TD-DFT/B3LYP has known limitations in describing certain excited-state properties, particularly charge-transfer and Rydberg states.

Nevertheless, B3LYP remains widely used in photochemical studies and provides a practical balance between accuracy and computational feasibility.

Moreover, to the best of our knowledge, TD-DFT is the only truly “black-box” electronic-structure approach currently capable of providing excited-state data for a broad range of molecular sizes and compositions across chemical space without requiring case-by-case adjustments to the calculation protocol. Higher-level methods such as computationally expensive EOM-CCSD approaches, while more accurate, have not yet provided data at the scale required for training a universal model, while multireference methods such as CASSCF or CASPT2 require system-specific settings, making them unsuitable for this task.

Furthermore, we note that OMNI-P2x is agnostic to the level of theory used and can be re-trained or fine-tuned to any other, potentially more accurate, method, with many examples shown in the manuscript. We also show that such transfer learning, even when starting from TD-DFT/B3LYP data, can greatly enhance the resulting model. This discussion has been added to the main text:

As with all machine-learning interatomic potentials, OMNI-P2x can only be as accurate as the reference method on which it is trained. In our case, this is TD-DFT/B3LYP/6-31+G*, which offers a practical balance between accuracy and computational feasibility across large chemical spaces, but is known to struggle with certain excited-state properties—most notably charge-transfer and Rydberg excitations. In small molecules, the use of a modest basis set such as 6-31+G* can further exacerbate this issue, as higher-lying excitations may correspond to poorly described Rydberg states. These limitations are intrinsic to the reference data and thus propagate into the trained model. Nevertheless, OMNI-P2x is fully agnostic to the level of theory, and can be re-trained or fine-tuned on more accurate data when available; in fact, we show in this work that transfer learning from TD-DFT/B3LYP already improves performance when adapted to higher-level reference methods.

Reviewer 2, Comment 4

Additionally, I found other major issues with the manuscript:

1. The only task for which I would recommend a trustworthy use of the model is the virtual screening of azobenzene derivatives, where being able to qualitatively and rapidly screen molecules can help identify and select potential candidates with desirable properties. Sadly, the quantitative performance shown in the UV/Vis test case is very poor. Even in the (potentially cherry-picked) example from the top quartile, the prediction is, at least partially, wrong. The oscillator strengths are off, and between 275 and 300 nm there are 4 states predicted vs 1 at TD-DFT/BLYP level. The authors claim an excellent agreement, probably referring to the position of the maxima, but anyway, the convolution of the spectra is highly affected by the artificial broadening of 0.3 eV, which might make the results look more in agreement than they actually are (additionally, the reference to justify 0.3 it is unrelated, as in this manuscript the error is between TD-DFT data and their prediction, and not a discrepancy between computed/computed at different levels or computed/experimental spectra).

We respectfully disagree with the remark that only trustworthy use of the model is the virtual screening of azobenzene derivatives. Before addressing the comment about spectra, we must point out that the fine-tuning of the model for surface-hopping dynamics is another clear example of the trustworthy use of the model. As for the UV/Vis spectra, we do agree with you that comparing spectral lines may introduce a bias: however, we believe that such a comparison is also important, as those are the fundamental, observable quantities in experimental studies, as opposed to oscillator strengths and vertical transition energies. Single-point convolution is a common approach and the broadening factor of 0.3 eV is not arbitrary but was derived in the cited reference by fitting the single-point convolution (SPC) spectra with those obtained with the nuclear-ensemble approach (NEA) on a set of representative molecules. NEA is a better approach for simulating spectra, which we added to the manuscript as another trustworthy use case of our model as described replies to other comments.

To provide an alternative ranking of the spectra not based on the criticized similarity of the spectral shapes, we added an additional benchmark where we score the predictions by the mean absolute error of oscillator strengths and vertical transition energies and provide an analogous example in Figure S5 of the ESI, with the main text now reading:

To provide a benchmark of the predicted spectra that is unaffected by the choice of a broadening factor in the SPC construction, we also rank the spectra by a geometric mean of the excitation energy and oscillator

strength MAE, which is presented in Figure S5 of the ESI. Later, we also explore the application of OMNI-P2x for spectra calculations without relying on SPC approximation and the broadening factor.

Reviewer 2, Comment 5

2. The overall idea of the UV/Vis task is to predict the brightness of states in a specific order. This is not enough for this task; it is just a collection of energies with no correlation with their brightness. No information about the character of the states is present, and why a state is bright or not. The adiabatic order of the states (S_1 , S_2 , etc.) has no correlation with what states they actually are. Two molecules could have S_4 both bright for two completely different and uncorrelated reasons, while two very similar molecules could have S_1 and S_2 with inverted brightness just because, for example, $n\pi^$ and $\pi\pi^*$ states are inverted. So if there is no information on the mapping between excited states, predicting oscillator strength of adiabatic states is useless, if not wrong.*

We thank you for this insightful comment. We agree that OMNI-P2x, like any ML potential, does not provide explicit information on the orbital character of the ground or excited states (e.g., it cannot distinguish $\pi\pi^*$ from $n\pi^*$ transitions), nor can it resolve cases where state ordering differs between molecules. Instead, the model predicts excitation energies and oscillator strengths in the adiabatic ordering defined by the reference method. While this limits interpretability in terms of electronic structure, it is consistent with how UV/Vis spectra are typically constructed from ab initio calculations, where oscillator strengths are reported for adiabatic states without explicit orbital labeling. The utility of OMNI-P2x is therefore in reproducing excitation spectra (energies and intensities) efficiently and at scale, rather than in providing mechanistic orbital analysis.

Reviewer 2, Comment 6

3. In this logic, I also cannot understand, and it's not explained, why the highest accuracy is achieved for the S_0 excitations to S_1 and to higher-lying S_8 – S_{10} states. I cannot see any correlation in that, as there is no information on the character of the state in the training. I believe there might just be a high density of states in some energy range that biases the training process.

We note that, contrary to your comment, the $S_0 \rightarrow S_1$ excitation energies have consistently exhibited the lowest accuracy in both the original and revised models, so we are unsure where the claim of “the highest accuracy” originated. For higher-lying excited states, we agree that the higher density of closely spaced states may make their energies easier for the ML model to learn,

resulting in lower prediction errors.

Reviewer 2, Comment 7

4. In the dynamics part, the impact of the new model is limited. There are no particular advances with respect to previous molecule-specific training for nonadiabatic dynamics. The model is not transferable and universal because it needs fine-tuning at a different level of theory, coming back to the problem of single-molecule training. It could proven a computational speed-up, but not a paradigm shift in ML application in NAMD. Additionally, I wonder why the work previously published in Nat. Comms. by Axelrod et al. is not cited, where an example of transferable ML architecture for NAMD is actually reported. (<https://www.nature.com/articles/s41467-022-30999-w>). Also more recent example of transferable NN for excited states should be mentioned (<https://arxiv.org/abs/2502.12870>).

We acknowledge that our original version might not have clearly demonstrated the advantage of our model for NAMD. We have re-worked the “Efficient ML-driven nonadiabatic dynamics” section to better highlight the advantages offered by using a foundational ML model as a basis for training MLPs for nonadiabatic molecular dynamics. Our new computational experiment, shown in Figure 10, shows that transfer learning allows for obtaining stable and accurate TSH simulations with an order of magnitude (2% vs 20%) less data than training from scratch. This is not only a valuable computational speedup, but also a paradigm shift in how MLPs for nonadiabatic molecular dynamics are designed, by leveraging the chemical space information contained in a pre-trained model, and fundamentally changing how AL loops are designed for NAMD applications. The main text now reads:

To assess the improvement in data-efficiency that comes from using a pre-trained universal model, we compare the performance of the OMNI-P2x fine-tuned and MS-ANI trained from scratch on fractions of the full training set required to converge an AL procedure (35071 points, as reported in Ref. 57). Looking at the stability of the dynamics, calculated as the percentage of trajectories finishing without dissociation, shown in Figure 10a, the fine-tuned OMNI-P2x model can propagate stable dynamics given just 0.1% of the dataset, while dynamics propagated using the pure learning approach become consistently stable above 10% of the data. Furthermore, Figure 10b presents the predicted quantum yield of trans to cis photoisomerisation, as a function of the training set size. For the TL potentials, all models trained on above 2% of the data predict QYs within the reference range, up to a 95% confidence interval error bar. On

the other hand, the performance of the models trained from scratch is much less consistent, with outliers predicting incorrect quantum yields even at high fractions of the training set.

The above findings highlight that pre-training of OMNI-P2x on a vast dataset allows it to gain knowledge on both the chemical compound space and some information about conformational space, greatly reducing the amount of data needed for its fine-tuning. This makes OMNI-P2x a method of choice for propagating ML-accelerated surface hopping dynamics after data-efficient fine-tuning. A similar phenomenon has been reported in Refs. [58, 99] where a MLP pre-trained on ground-state data was able to achieve higher accuracy for excited-state predictions than an MLP trained from scratch, in the low-data regime.

We thank you for bringing our attention to above manuscripts. We have included citations to these works where appropriate (one of them in the passage above, with another relevant pre-print from the same group). We would also like to mention, that the underlying architecture of OMNI-P2x, the MS-ANI model was shown in a proof-of-concept experiment to potentially be able to perform NAMD simulations across chemical space, as shown in Ref. 57. The main text now reads:

This has changed with the MS-ANI (multi-state ANI) model [57] demonstrated to be able to predict nonadiabatic dynamics of three distinct molecules within the same model, paving the way for universal excited-state potentials. More recently, another demonstration of a possibility to create transferable NN for excited-state predictions was shown in Ref. 58. Transferable nonadiabatic molecular dynamics simulations were also reported by Axelrod et al. [59] however, it was limited to the class of diazobenzene photoswitches.

Reviewer 2, Comment 8

5. The fulvene dynamics look very different from the reference in literature, where a reflection in the S_1 population is present, particularly with trajectory surface hopping (Phys. Chem. Chem. Phys., 2020,22, 15183-15196). This is due to a physical reason, with the molecule encountering two different conical intersections, one of them populating back the first excited states. This is the reason why this model has been chosen to test nonadiabatic methods. Already the reference method used in this manuscript misses this effect. The training with the model actually fully deletes the small reflection at around 15 fs, removing completely the physical behavior expected. I believe this is more due to interpolation reasons than capturing the complexity of the potential energy surface, but in general is an example where the prediction erases the correct physical behavior and potentially leads to misinterpretation. This could be solved using another formulation of TSH, for which nonadiabatic coupling or wavefunction overlap would need to be predicted by the model, but is not available in the architecture presented in this work.

Thank you for bringing our attention to this. We would like to point that the reflection populating back the excited state is present in the reference dynamics, however, its magnitude is indeed smaller than in the FSSH dynamics. Furthermore, the fine-tuned ML model captures this physical behavior significantly better than the no-TL model, trained on the same dataset. To emphasize this, we have included an inset in Figure 9, zooming in on the 10 fs to 20 fs region of the simulated populations. We have added 95% confidence interval error bars showing the statistically significant difference between the MLIPs trained from scratch, and using transfer learning, while also increasing the number of trajectories propagated using both ML approaches to 10,000, as shown in the new Figure 9.

The text accompanying this figure now reads:

By employing transfer learning from a universal potential, we are able to slash the convergence time of fulvene from over three days, to about 24 hours, with a ca. three-fold increase in computational efficiency. Additionally, the final employed training set contains only 3550 conformations, which is about twice less as was necessary to achieve convergence in a reference direct active learning procedure, as can be seen in Figure 9. Moreover, inspecting the time evolution of the electronic state populations, we can see a statistically significant difference in the populations predicted by the fine-tuned model and the previous MS-ANI model. Despite a smaller training set, the TL version of OMNI-P2x better captures the photophysical behavior of the system.

The agreement between the FSSH and LZSH methods of propagating NAMD is greater if the reduced kinetic energy reservoir option is disabled. A comparison of populations obtained by propagating TSH with three different approaches: ML-LZBL (no reduced kinetic energy), ref-LZBL (no reduced kinetic energy) and FSSH is provided as Figure S12 in the ESI, showing that the LZBL surface hopping scheme can accurately reproduce FSSH. However, as the main goal of designing ML potentials is replicating

the underlying reference calculations, we choose to keep the original comparison, as that was the computational setup used in the active learning procedure.

Additionally, a comparison of the dynamics predicted using different ML models and different LZBL propagation settings with reference fewest switches surface hopping dynamics is available in Section S4 of the ESI.

Reviewer 2, Comment 9

6. More information about the computational cost of training would be necessary. Yes, the prediction is definitely cheaper than TD-DFTB, but working on a large dataset, what is the computational cost of the training with respect to calculating these excitations with the semi-empirical method?

As OMNI-P2x is a foundational model, it can be used to make predictions on any dataset without further training; hence, we believe it is fair to compare the model inference time with the prediction speed of the semi-empirical methods. In the same spirit, semi-empirical methods require time (often years, as in the case of ODM2) to parametrize, which is not included in the total computational cost of individual calculations, once the method is already developed and parametrized. However, we agree with the referee that including training times and hardware information is beneficial to the clarity and fairness of the manuscript; hence, we include an additional “Training times” section in the ESI, with this information provided.

The 4 main models of OMNI-P2x (ensemble of three energy-predicting NNs and oscillator strength NN) have been trained for 100 epochs on 32 NVIDIA GH200 GPUs with 96GB memory each, which totaled to about 48 hours of training time per model. Fine-tuning of OMNI-P2x for NEA spectra was performed in a Jupyter Lab environment, with 4 Intel Xeon Gold 6226R CPU cores, taking under a minute for the pyrazine case, and about 40 minutes for 9-DCMA. The AL loop for NAMD simulations of fulvene was run on a computational node with a single NVIDIA 4090 RTX GPU and 16 Intel Xeon Gold 6226R CPUs, taking 24 h to converge. Fine-tuning of OMNI-P2x for NAMD simulations was performed with the same computational resources, taking about 20 minutes per trained model.

Minor points:

Reviewer 2, Comment 9

Is the label of Figure 4a correct? I think the authors show the ground state energies.

Thank you for your vigilance. This caption was indeed a typo, and was changed to read:

Test performance of the OMNI-P2x model. Panel (a): correlation plot of

the reference~(true) and OMNI-P2x predicted ground-state energies at the CCSD(T) level of theory, for the test part of the ANI-1CCx dataset; (b) MAE of the electronic state energies; c) MAE of the excitation energies; d) MAE of the predicted oscillator strengths.

Reviewer 2, Comment 10

The statement “While these methods are certainly more affordable for systems with hundreds and even thousands of atoms, the accuracy of their predictions can often be questionable” needs a reference.

We thank you for their comment. References have been provided to justify that claim:

While these methods are certainly more affordable for systems with hundreds and even thousands of atoms, the accuracy of their predictions can often be questionable. [26–28]

Reviewer 2, Comment 11

Some self-glorifying strong claims like “This marks the watershed moment that ML approach for excited-states presents a viable alternative to established QM methods.” should be toned down, as they just represent the author's personal opinions.

This sentence has been re-formulated to read:... making ML approaches for excited-states a viable alternative to established QM methods.

Reviewer #3

This communication presents a novel foundation model / universal model for the excited state potential energy surfaces of organic molecules. The introduction motivates the model well in terms of deficit of the capabilities of existing computational methods for excited states in combination with high-throughput or dynamics-based studies of anything beyond very small molecular systems. The approach taken is to adapt a variant of the well-established ANI neural network potential, recently-introduced by the current authors and other collaborators in [54] npj Computational Materials 11, 132, which introduces a state-index as an extra input to the neural network alongside the geometry and atomic numbers, resulting in a “MS-ANI” approach. The current work builds on the demonstrations in [54] by showing that a foundation model, OMNI-P2X, can be built with this architecture which can be general across a fairly wide range of organic molecules. Overall the model is demonstrated to be quite successful and I think the paper could be quite influential as it represents a first of its kind. However, I feel there is room for improvement in the presentation and demonstrations before publication: in several sections the results themselves are very briefly described and it is hard to discern the level of success of the model.

Thank you for the positive evaluation. In the presented revision, we have taken numerous steps to increase the accuracy of the OMNI-P2x model by doubling the training set size, increasing the computational effort used in the training process, and employing an ensemble of NN models. Furthermore, the manuscript has been revised to present more clearly the advantages and limitations of OMNI-P2x. An additional workflow for fine-tuning the model to predict spectra based on non-equilibrium structures using the nuclear ensemble approach has been added in a new section, “Nuclear ensemble approach spectra”.

Reviewer 3, Comment 1

Training data has been sourced from two existing databases, the PubChemQC database of TD-B3LYP/6-31+G calculations, and the ANI-1ccx dataset of small molecules including off-equilibrium configurations, which has been provided as ground and excited-state DFT and TDDFT data for 10 excitations, and as ground-state-only CCSD(T), allowing the model to predict multiple levels of theory. Results show the S0 ground state has an MAE on energies of 2.63 kcal/mol which makes it comparable to other models trained on large databases of organic molecules, though I would prefer to see an explicit comparison to understand clearly how good the model is for the ground state.*

We agree that such a comparison would be beneficial. However, ground state was not the main focus of this study, and better results can be obtained with more ground-state training data (we used a quite small data set compared to other universal ground-state potentials). Comparison to other MLIPs for ground-state properties has been added to the main text, comparing the ground-state predictions at the DFT level:

This ground-state MAE can be put in perspective to the similar metrics of other MLIPs: ANI-1 with an RMSE of 1.3 kcal/mol (OMNI-P2x 2.93 kcal/mol), UMA [76] with around 1 to 3 meV/atom, depending on the version (OMNI-P2x 4.48 meV/atom), and AIMNet2 [77], reporting a MAE of around 1.25 kcal/mol for neutral compounds. Although those results are not directly comparable, as each of the models was trained on a different dataset, they demonstrate that the ground-state error of OMNI-P2x exhibits decent performance, even with respect to the current state-of-the-art models, despite being trained on much smaller ground-state data, as ground-state performance was not the primary focus of this study.

Reviewer 3, Comment 2

The S1 state and higher-lying excitations are substantially worse, at ~5 kcal/mol, which is still useful, but it would be interesting to understand how correlated are the errors between S0 and the other states, ie should we expect errors on S0->Sx transition energies to exceed those of the contributing states, or will there be error cancellation to some degree? Scrutiny of Figs 4b and 4c might provide this information in an averaged sense, but they have different energy units so the comparison is not straightforward (generally, I find the broad selection of energy units (variously Ha, eV, and kcal/mol, sometimes in the same figure) rather troubling.

Thank you for your insightful questions. The units used in Figure 4 have been unified to kcal/mol. Furthermore, the version of the model presented in the

manuscript has been revised, replacing the original OMNI-P2x with an ensemble of three neural networks, newly trained on the entirety of the PubChemQC dataset. The ensemble outperforms the previous version of the model by about 30% in terms of the MAEs. Furthermore, a detailed breakdown of the performance of OMNI-P2x predictions of vertical excitation energies has been provided in the ESI, featuring correlation plots with R^2 metrics and MAE for all 10 transitions. The excitation energy errors are similar in magnitude to the target state energy errors and follow the same trend: largest for the $S_0 \rightarrow S_1$ transition and decreasing for higher-lying transitions.

Reviewer 3, Comment 3

Figure 3 shows the distribution of sizes and excitation energies of the training data, indicating that the majority of the PubChemQC data is in the 20-40 atom range whereas that of the ANI-1ccx data is much smaller, peaking at 12 atoms. Figure 4 shows some test results: I note that while figure 4a purports to demonstrate close agreement between predicted and target total energy, the spread of energies is so wide (1000 Hartrees!) that presenting this as a $y=x$ figure is highly redundant – any deviation from $y=x$ visible at this scale would represent an enormous error, and the MAE of 5.78kcal/mol \sim 0.01 Hartree would be invisible on this scale. Perhaps a baseline subtraction of the atomic energies from each case would make this graph more useful?

Thank you for this suggestion on improving the presentation of our results. Panel a of Figure 4 has been re-worked to show the MAE error on normalized energies (with removed self-atomic energies), and in the main text, we report the MAE, as well as the correlation of the normalized and un-normalized energies:

For the ground-state energies in the test ANI-1ccx subset, OMNI-P2x has a higher MAE of 4.14 kcal/mol, but still excellent $R^2 = 0.9957$ for the normalized energies (0.9999 for unnormalized).

Reviewer 3, Comment 4

Regarding the fact that 10 excited states are calculated: are this many states always going to give meaningful results in these systems with a 6-31+G basis: in small molecules these might be higher-lying Rydberg states which are not well-represented in a relatively small basis such as 6-31+G*?*

We thank you for this comment. We agree that for smaller molecules, TD-DFT with a modest orbital basis set such as 6-31+G* may, in certain cases, yield unphysical results for higher-energy transitions. However, selecting a larger basis would lead to prohibitively expensive calculations, considering a training set consisting of over 3k medium-sized molecules. It is also unfeasible to define

one excitation-number cutoff limit that would exclude Rydberg states from the training set, without manual individual, manual inspection. At the end of the day, the applicability of all ML models can never go beyond the applicability of the underlying reference method, and the user should carefully judge if the best-case scenario: exact replication of TD-B3LYP/6-31+G* results is suitable for their study. The manuscript has been updated with that information:

As with all machine-learning interatomic potentials, OMNI-P2x can only be as accurate as the reference method on which it is trained. In our case, this is TD-DFT/B3LYP/6-31+G*, which offers a practical balance between accuracy and computational feasibility across large chemical spaces, but is known to struggle with certain excited-state properties—most notably charge-transfer and Rydberg excitations. In small molecules, the use of a modest basis set such as 6-31+G* can further exacerbate this issue, as higher-lying excitations may correspond to poorly described Rydberg states. These limitations are intrinsic to the reference data and thus propagate into the trained model. Nevertheless, OMNI-P2x is fully agnostic to the level of theory, and can be re-trained or fine-tuned on more accurate data when available; in fact, we show in this work that transfer learning from TD-DFT/B3LYP already improves performance when adapted to higher-level reference methods.

Reviewer 3, Comment 5

It is interesting that the Energy MAE on S1 is roughly double that of S0, then further higher-lying excitations have increasingly lower errors. It strikes me that perhaps the increasing Rydberg character of higher-lying states is somehow easier for the model to capture. The authors say “An outlier here is S0, which has the smallest energy MAE of 2.63 kcal/mol, which we attribute to the effect of the added ground-state CCSD(T) data” – I couldn’t see how that could come about – why would the presence of data associated with a second level of theory predictable by the same model improve the accuracy of its predictions for the first?

We are thankful for this insightful comment. Regarding the prediction accuracy of higher excited states, it is indeed possible that their character makes machine learning fitting easier. In particular, the density of states (i.e., the number of states per energy unit) increases with excitation number; if the model captures this trend, it can lead to lower MAEs for higher-lying states.

Concerning the improved accuracy for S₀ when training on multiple levels of theory, this phenomenon has been discussed in the all-in-one learning (AIO) framework (ref. 63, 10.1021/acs.jctc.5c00858.). AIO-trained models often show

similar or even larger improvements compared to transfer learning models pre-trained on DFT data and fine-tuned to coupled-cluster quality. Specifically, transfer learning from ground-state data has been shown to enhance excited-state property predictions (ref. 58, 99), and we expect AIO to yield analogous benefits here. At the same time, the dramatic improvement observed for S_0 , roughly a twofold decrease in MAE, likely also reflects characteristics of the training data itself, as ground-state energies generally vary more smoothly with molecular geometry, exhibit lower variance, and follow distributions that are easier for the model to learn.

The manuscript has been updated to read:

An outlier here is S_0 , which we attribute to the improved data-efficiency due to the application of the all-in-one learning to the combined set with the ground-state DFT (from PubChem) and the CCSD(T) (from ANI-1ccx) data, and an overall smoother and lower-variance distribution of ground-state energies that makes them easier for the model to learn.

Reviewer 3, Comment 6

Some minor points on figures: “One can note, that the highest accuracy is achieved for the S_0 excitations to S_1 and to higher-lying S_8 – S_{10} states.” Based on Figure 4d this does not appear to be true – do they mean the lowest accuracy?

Thank you for catching this mistake. As the entire OMNI-P2x model was revised and re-trained, the sentence was changed to reflect the true accuracy of the predicted oscillator strengths:

One can note that the highest accuracy in terms of the predicted oscillator strengths is achieved for the excitations to S_1 and to higher-lying S_8 – S_{10} states.

Reviewer 3, Comment 7

In the stick plot in Figure 5: the baseline of sticks does not align with zero on right hand axis – this needs correcting.

Thank you for noticing this: the plots have been aligned so that the oscillator strength axis aligns with the bottom of the stick spectrum.

Reviewer 3, Comment 8

Panel 8a is very separate from 8b,8c: they refer to different systems and unrelated calculations.

We agree with you on this matter. We have now separated Figure 8 into Figures 9 and 10, with a separate figure showing the results of the active learning loop

for fulvene, and a separate figure showing the usage of OMNI-P2x in transfer learning models for simulating the photoisomerization dynamics of azobenzene. Further discussion of these plots follows in a response to a question further in this letter.

Reviewer 3, Comment 9

I would say that the Statement “OMNI-P2x is approaching the accuracy of time-dependent density functional theory” in the abstract and main text is potentially misleading, despite being carefully written. As Figures 5 and 6 show, while the model does significantly better than semi-empirical methods, at reproducing TDDFT spectra, as measured by the Spearman Correlation Coefficient, the reasons for this in terms of agreement of the underlying stick spectrum with TDDFT are often dubious. In many cases, where TDDFT has a few strong peaks, OMNI-P2X has a forest of intermediate peaks in a similar energy range, with no clear relationship between oscillator strengths. I would want to know what the Mean Relative Error on oscillator strengths was, as well as the MAE, because it seems like in many cases it might be quite large. For a method to be able “approach the accuracy of TDDFT” I would expect individual excitations to be predicted to better than the 5kcal/mol quoted in Figure 4b as the MAE on S1 energies, and for the mean relative error on oscillator strengths not to be too extreme. This current level of energy error corresponds to 0.22eV or 100nm on a S0<-S1 excitation around 1.5eV, which is not really approaching TDDFT accuracy.

Thank you for pointing this out. We agree that the accuracy of predicting spectral lines may not necessarily be correlated with the accuracy of the predicted underlying stick spectrum. To quantify this, we reproduce Figure 7 using the total error of the absorption energies and oscillator strengths as a ranking, instead of the Spearman Correlation Coefficient. The results are available in Figure S5 of the ESI.

We also agree that the level of error presented in the original model, of 0.22 eV, is rather large, yet it was still comparable to the typical error of TD-DFT predictions of 0.2-0.3 eV. However, the revised ensemble of NNs is able to achieve a lower error, below 4.0 kcal/mol (0.17 eV) for all states, making the error related to fitting the ML model lower than the inherent error of TD-DFT. Furthermore, a more thorough comparison is available in Figure S4 of the ESI, showing correlation plots for excitation energy predictions for all 10 transitions. The main text now reads:

Furthermore, the MAEs for both the predicted total and excitation energies are below 0.2 eV (4.6 kcal/mol), which lies below the lower bound of the reported accuracy for the reference TD-DFT method, indicating that the

fitting error of the ML model is smaller than the intrinsic error of the underlying quantum chemical approach. Correlation plots showing the predicted vs true excitation energies for all transitions are available as Figure S4 of the ESI. In all cases, the R^2 is above 0.91.

Reviewer 3, Comment 10

The section on design and screening of azobenzene compounds provides a proof-of-principle for use in screening large datasets of candidate molecules. In this case azobenzene derivatives are screened to find 21 leads with strongly red-shifted absorption, at least some of which stood up to further scrutiny via calculations at a higher level of theory, ADC(2). I found it frustrating that this section ended without serious investigation of whether OMNI-P2X had really found strong leads: no comparison is made of the spectra from TD-DFT, from ADC(2) or from OMNI-P2X. I would want to see this investigated more deeply.

We agree with you on that. Indeed, a more thorough comparison of the different levels of theory used to screen the azobenzene derivatives would be beneficial. Following this suggestion, we have now included the SPC spectra for the top 3 candidates presented in Figure 7 in the Electronic Supporting Information. Overall, we note a good level of agreement between the methods, with S_1 and S_2 excitation energies' MAEs of about 0.2 eV between OMNI-P2x and TD-DFT and 0.42 and 0.35 eV between TD-DFT and ADC2 and OMNI-P2x and ADC2, respectively. Furthermore, molecular databases containing molecules with state energies at the three discussed levels of theory are provided, along with a Jupyter notebook and instructions on how to generate and compare spectra for any given compound.

Reviewer 3, Comment 11

The final section demonstrates what could turn out to be a common use-case for foundation models for the excited state: active-learning-driven fine-tuning / transfer-learning is used to train a OMNI-P2X model at the CASSCF(6,6) level of theory for fulvene. The transfer-learning is shown to substantially accelerate model training, and both the TL and no-TL versions accurately reproduce dynamics of the reference CASSCF calculations. Very brief mention is made of calculations on photoisomerisation of azobenzene, but the results presented are rather cryptic: I was not able to intuit what point was being made about dihedral angles as obtained from the dynamics.

We agree with your observation and now substantially expand the section concerning the nonadiabatic molecular dynamics, with particular focus set on the example of azobenzene. The results for fulvene have been separated (Figure 9) from the azobenzene results, presented in Figure 10. We also added a zoom-in inset on the crucial state-population region between 10 and 20 fs of the dynamics, which clearly shows the difference in populations predicted by the TL and no-TL models, with the version derived from OMNI-P2x predicting quantitatively accurate dynamics, in agreement with reference populations. Additionally, shaded 95% confidence interval error bars have been added to

show the statistical significance of the results. The main text now reads:

By employing transfer learning from a universal potential, we are able to slash the convergence time of fulvene from over three days, to about 24 hours, with a ca. three-fold increase in computational efficiency. Additionally, the final employed training set contains only 3550 conformations, which is about twice less as was necessary to achieve convergence in a reference direct active learning procedure, as can be seen in Figure 9. Moreover, inspecting the time evolution of the electronic state populations, we can see a statistically significant difference in the populations predicted by the fine-tuned model and the previous MS-ANI model. Despite a smaller training set, the TL version of OMNI-P2x better captures the photophysical behavior of the system. Additionally, a comparison of the dynamics predicted using different ML models and different LZBL propagation settings with reference fewest switches surface hopping dynamics is available in Section S4 of the ESI.

To assess the improvement in data-efficiency that comes from using a pre-trained universal model, we compare the performance of the OMNI-P2x fine-tuned and MS-ANI trained from scratch on fractions of the full training set required to converge an AL procedure (35071 points, as reported in Ref. 57). Looking at the stability of the dynamics, calculated as the percentage of trajectories finishing without dissociation, shown in

Figure 10a, the fine-tuned OMNI-P2x model can propagate stable dynamics given just 0.1% of the dataset, while dynamics propagated using the pure learning approach become consistently stable above 10% of the data. Furthermore, Figure 10b presents the predicted quantum yield of trans to cis photoisomerization, as a function of the training set size. For the TL potentials, all models trained on above 2% of the data predict QYs within the reference range, up to a 95% confidence interval error bar. On the other hand, the performance of the models trained from scratch is much less consistent, with outliers predicting incorrect quantum yields even at high fractions of the training set.

The above findings highlight that pre-training of OMNI-P2x on a vast dataset allows it to gain knowledge on both the chemical compound space and some information about conformational space, greatly reducing the amount of data needed for its fine-tuning. A similar phenomenon has been reported in Refs. [58, 99], where an MLP pre-trained on ground-state data was able to achieve a lower MAE for excited-state predictions than an MLP trained from scratch, in the low-data regime.

Figure 10 now shows a new computational experiment demonstrating the improvements associated with using OMNI-P2x as a pre-trained model for constructing MLIPs for propagating trajectory surface hopping simulations. Namely, we measure the stability of the conducted simulations, as well as the predicted quantum yields, as a fraction of the full AL dataset from Ref. 57, for TL and no-TL models. The results show that OMNI-P2x can perform stable simulations (without any dissociated trajectories) with as little as 0.1% of the full training set and achieves quantitative agreement with the reference predicted quantum yields with orders of magnitude less training data than ML models trained from scratch.

Reviewer 3, Comment 12

One natural question would be how the model responds to large molecular systems and relatively large numbers of states required, and is there consistency of the predictions with respect to the number of states requested. In some testing of the version made available on figshare, I observed that when calling OMNI-PX2 three times, for 2, 5 and 10 states, for a model of a sunscreen candidate molecule with 95 atoms, in each case I got a very different set of excited states predicted (the S1->S0 transition was much lower in the 10 state case than the 2 state case). Is there anything that guarantees consistency with respect to the number of states requested. I note it is relatively stable for the provided example of benzene, though the oscillator strength varies a little.

We thank you for the question. The OMNI-PX2 model is designed as a multi-state model, in which predictions for each electronic state are made consecutively (i.e., with the neural network called for $x = 0, 1, 2 \dots$ up to the requested number of states). As a result, the predicted energies are, in principle, independent of the number of states requested.

We were able to reproduce this effect using Bemotrizinol (a 95-atom sunscreen molecule). In this case, the predicted energies and oscillator strengths remain constant; however, because the molecule is significantly larger than the molecules in the training set (median ~ 30 atoms, maximum 65 atoms), the ordering of higher electronic states may differ from the physically correct order. This can lead to apparent lower-energy peaks in the simulated spectrum, even though the model's predictions for individual states are consistent.

The revised release of OMNI-PX2 addresses such situations through an uncertainty quantification system, which issues warnings when predictions are highly uncertain, potentially due to the choice of a system far from the training regime.

The manuscript has been updated to clearly state:

Furthermore, the neural network model is expected to produce reliable predictions primarily for molecules whose size and chemical composition are similar to those represented in the training set; extrapolation to significantly larger or chemically distinct molecules may lead to unphysical or unexpected results. The built-in uncertainty quantification mechanism warns the user about predictions that are not covered well by the training set.

Responses to the comments of the reviewers

Reviewer comments are in *italics*, and our responses are in **regular, blue font**. Our changes to the manuscript are marked in **red font**.

Reviewer #1

Thank you very much for the detailed reply to my comments. After the revision, I think the limitations are more openly communicated and indicated in the manuscript. As the manuscript is certainly a step forward towards generalizable prediction of excited state energies, I support the publication in Nature Communications..

We thank you for your positive opinion of the manuscript and your thought-provoking comments, which led to substantial improvements.

Dimers: Thank you for the additional experiments. The fragment-correction scheme is very interesting and a valuable addition to the manuscript.

Level of theory encoding and excited states encoding: Thanks for the additional ablation study.

Out-of-equilibrium: Thanks for the very interesting additional tests, particularly Figure S10 and the fine-tuning study.

Figure 4: Thanks for the changes.

Figure 6: It is a very interesting observation that out-of-equilibrium errors are higher for lower excited states. Thanks for adding this information.

Delta-Learning: I understand the complexity of this topic, and I am looking forward to seeing a more detailed study of this. I agree that it might warrant a separate paper.

Transfer learning: Thank you for making this clearer and adding the new figure to indicate the impact of transfer learning.

Reviewer #2

I thank the authors for addressing my concerns. I acknowledge the value of the new section on nuclear ensemble approach spectra, the addition of the nonadiabatic dynamics (NAMD) of azobenzene, and the discussion of some of the limitations of the model. I believe that the revision improved the manuscript.

Thank you for your careful review, which led to the manuscript improvement.

However, I believe that my chemistry-related comments have not been sufficiently scientifically addressed. Especially, if the model has already received high visibility as the authors report, it is even more important to ensure that it delivers physically correct results.

Below, we address the remaining chemistry-related comments and hope that this revision will find your approval.

Additionally, I suggested some potential modifications that might make the impact of the model strong enough to recommend publication, like the prediction of the character of the excited states, nonadiabatic coupling, or spin-orbit coupling. However, none of these have been addressed, although in some cases, even acknowledged that architecture is inherently extensible for this. That was not only required to show a justification of the novelty of the model to be published in Nat. Comm., but also to avoid potential flaws in the chemical interpretation that can be obtained by using the model as it is.

We acknowledge that interpretations are as important as massive simulations, which our model enables. We agree with you that elucidating the character of the excitations is one of the most crucial tasks in theoretical modeling of excited-state processes. Hence, in the revised manuscript, we show that the OMNI-P2x model provides a unique insight into the character of excitations.

This insight is based on an analysis of the atom-wise contributions to excitation energies, which are readily obtained from the OMNI-P2x predictions. The OMNI-P2x-derived atom-wise contributions to excitation energies agree well with the density difference plots from the reference TD-DFT calculations. Hence, in the revised manuscript, we have introduced a new interpretable machine learning approach for excited states.

The new section of the revised manuscript provides the details of this new approach:

To provide interpretability of the ML-learned excitations, we introduce a new approach for analyzing atom-wise contributions to excitation energies. For each atom i , the fractional contribution is defined as $|\Delta E_i|/\sum|\Delta E_i|$, where ΔE_i is the excitation energy contribution for a given transition from atom i , calculated as the difference between ground- and excited-state atomic energies predicted by the NN ensemble. We show

that the OMNI-P2x-derived atom-wise contributions agree well with the reference TD-DFT density difference plots (Fig. 8). Such analysis allows us to obtain the spatial character of the excitation by identifying which atoms are most involved in the transition. Our analysis is conceptually related to work by Tkatchenko et al [83], analyzing atom-wise contributions to ground-state energies.

We demonstrate that this interpretability analysis can work well across chemical space and for different excitations of the same molecule. Panel a) of Fig. 8 shows the $S_0 \rightarrow S_1$ ML ΔE contributions and TD-DFT density differences for several organic molecules. We begin by comparing the examples of benzene and pyrazine, two 6- π electron aromatic compounds. OMNI-P2x correctly localizes the transitions, spreading it evenly across all the benzene carbon atoms, while in pyrazine it is mostly localized on the two nitrogen atoms, reproducing the TD-DFT results. The excitations also exhibit appropriate symmetry: C_6 for benzene, and D_{2h} for pyrazine. The next example, azobenzene, shows physically correct localization of the excitation on the N=N double bond, while in *p*-nitroaniline the partial charge-transfer character

of the excited state is well reproduced. Overall, this shows an interpretable character of ML predictions, that is transferable across chemical space.

Atomic contributions to different electronic transitions of the same molecule are shown in Fig. 8b), using azobenzene as an example. OMNI-P2x correctly reproduces the $S_0 \rightarrow S_1$ transition as localized on the central N=N bond, while the S_2 transition has larger contributions from the aromatic rings. Finally, a higher excited state (S_{11}) is shown, which is delocalized across the entire conjugated system.

I would like to justify my claims about the problems that the authors addressed and that persist in this revised version of the manuscripts.

We thank you for acknowledging the improvements in the manuscript, and the expansion of the NAMD results of azobenzene. We would like to clarify, that by “extensible”, we meant the extensibility to charged systems, and systems of other multiplicity, which were mentioned in your original comment, as stated in the manuscript: “Future extensions of the framework will aim to predict not only for neutral molecules in the singlet manifold, but also for states of higher multiplicity and charged species. Although the present focus on singlet states already enables accurate and chemically diverse predictions for many photophysical applications, the architecture is inherently extensible, and these additional capabilities will be the subject of future work.” As for SOCs and NACs, we make no such claims. The problem of predicting couplings that are widely transferable across chemical space remains an open problem. None of the reported models that incorporate non-adiabatic couplings can predict them across chemical space, and the feasibility of such a universal model is unknown. We would like to reiterate that by “universal potential”, we use the common definition signifying a model that has been pre-trained on a vast region of chemical space, and can be used out of the box for unseen compounds, not a

model that can universally predict every relevant quantity that can be obtained with quantum chemistry. The term universal has been applied to many models that predict only the ground-state energy, and its derivatives (eg. ANI-1, MACE, UMA). To avoid misleading anyone, a definition of a universal MLP is included in the manuscript:

A particularly important, emerging class of potentials is universal MLPs, which we define as models that were trained on vast datasets spanning across chemical space. In turn, they can be used to make out-of-the-box predictions for previously unseen molecules, generalizing from the training data.

The authors say:

“However, many (if not most) machine learned interatomic potentials (MLIPs) that are considered universal are limited to a certain set of elements, larger or smaller than OMNI-P2x.”

I agree with the authors. However, not clearly going beyond the state-of-the-art in this regard, it shows indeed the limited impact of the model in terms of offering a new paradigm for using MLIP for computational chemistry predictions.

We respectfully disagree with the comment that OMNI-P2x does not go beyond state of the art. OMNI-P2x goes beyond the current state of the art for MLPs by providing predictions for excited states, which no universal MLP has done before. This is a new paradigm in ML-assisted computational chemistry, while inclusion of more elements, while valuable, is an incremental change that will be the subject of further work, in the same manner as it is done for ground-state models.

My concerns about the correctness of the results used for training the model have not been sufficiently addressed or disproved.

The authors say: “The choice of reference method is obviously important and we fully acknowledge that TD-DFT/B3LYP has known limitations in describing certain excited-state properties, particularly charge-transfer and Rydberg states. Nevertheless, B3LYP remains widely used in photochemical studies and provides a practical balance between accuracy and computational feasibility.”

First of all, I disagree with the statement. Although it could reproduce good energies for some bright states, the danger of using B3LYP in excited states calculations over its benefits is very well established. That’s why several range-separated hybrids, or functionals with higher percentages of Hartree-Fock exchange, are strongly recommended to be used for excited states calculations, and they do not add any relevant computational expense. However, the widely used approach in literature does not justify its correctness, especially because the performances of density functional approximations are highly case-specific. It would have been necessary to prove the accuracy of B3LYP with higher reference calculations or a systematic evaluation of the performance. This would have been very easy to show, for at least a pool of selected molecules, but it has not been done. Unfortunately, acknowledging the limitations of B3LYP, but saying that it is highly used, without showing a rigorous evaluation of its prediction, does not support the claim supported by scientific evidence and limits the trustworthiness of the accuracy of these calculations that are used to train the model and consequently its reliability to be used for unseen molecules.

However, I thank the author for addressing this problem in the discussion.

We agree with the need to provide additional information and benchmarks for the underlying quantum chemistry method of OMNI-P2x. Fortunately, a great many literature benchmarks of B3LYP performance have been established in

the past. Consistently, B3LYP has emerged as one of the better performing functionals, with MAE errors ranging between 0.2 and 0.4 eV, with respect to experiment (Refs. 101 and 102) or a theoretical best estimate (TBE) (Refs. 103-106). We supplement this result with our own benchmark, comparing TD-DFT/B3LYP results with energies obtained with ADC(2), which is a higher-level method, suggested by you in the review, with results provided below:

Overall, there is a very good correlation between the ADC(2) results and B3LYP, with a correlation coefficient of about 0.95 for all electronic states. The MAE ranges from about 0.24 eV for $S_0 \rightarrow S_1$ transitions, up to 0.59 eV for $S_0 \rightarrow S_{11}$. However, it is worth noting the mean signed error of each transition, which is negative and in almost all cases equal to approximately the MAE value, which indicates a systematic underestimation compared to ADC(2), rather than random errors. Full correlation statistics have been presented in Figure S19 and Table S3 of the ESI.

In terms of oscillator strengths, benchmark studies by Thiel et al [106] show good correlation between B3LYP and CC2 results, with a MAE of 0.06, with accurate predictions of transition dipole moments demonstrated by Robinson et al on a smaller test set [109].

Furthermore, it should be noted that benchmarks comparing ADC(2) and B3LYP results show a somewhat better performance of ADC(2) when predicting excitation energies (0.1 vs 0.2 eV MAE in Ref. 103, 0.12 vs 0.30 in Ref. 108, and 0.21 vs 0.45 Ref. 105), the benchmark of Sarkar et al [105] shows that ADC(2) and TD-DFT/B3LYP predict oscillator strengths of about the same accuracy, with an error of 15.4% and 17.6% respectively, when compared to TBE. Hence, we do not compare oscillator strengths predicted by these two methods, as they are at the same level of error in this regard.

A discussion on the accuracy of B3LYP and cited benchmarks has been added to the manuscript:

As such, the expected level of error of TD-DFT/B3LYP, which is usually between 0.2 and 0.4 eV with respect to experimental benchmarks[103,104], or theoretical best estimates (TBE) [105-108], will be retained. The same is true for the oscillator strengths, which have reported MAE of between 0.05 and 0.15, compared to TBE [105,109]. We supplement these benchmarks by providing a comparison between ADC(2) and TD-DFT/B3LYP vertical excitation energies in Fig. S16 and Table S3 of the ESI. The DFT results correlate well with ADC(2), showing a systematic underestimation of excitation energies, which increases when moving to higher excited states.

Furthermore, we believe that OMNI-P2x opens the possibility to use higher levels of theory for complicated tasks *via* efficient fine-tuning, as demonstrated on the example of nuclear ensemble approach spectra or nonadiabatic molecular

dynamics, reducing the number of points that need to be labeled by orders of magnitude.

Additionally, the authors claim: “Moreover, to the best of our knowledge, TD-DFT is the only truly “black-box” electronic-structure approach currently capable of providing excited-state data for a broad range of molecular sizes and compositions across chemical space without requiring case-by-case adjustments to the calculation protocol.”. This is not true. ADC(2), available in several software (ORCA, TURBOMOLE, Q-CHEM) is black-box and more accurate. Its error is well-known, it does not require any case-by-case adjustment, and it does not depend on the mathematical form of the functionals chosen. It is yes, more expensive, but I believe affordable for the pool of molecules used in this work.

You are right that beyond TD-DFT, there are other ‘black-box’ methods. Thank you for suggesting the ADC(2) method. We agree with the assessment that ADC(2) is a black box method that can provide more accurate predictions of spectra for organic molecules. However, ADC(2) is prohibitively expensive for data generation at the scale presented in the manuscript with our resources: The generation of 1000 electronic spectra for the benchmark presented in this response letter required over 25000 CPU hours (using 12 AMD EPYC 9654 CPU cores with the TURBOMOLE program, as suggested). By extrapolation, covering the entire training set would require over 76 million CPU hours, almost two orders of magnitude higher than the cost of TD-DFT, which is unfortunately, not feasible for us.

Regarding the UV/Vis spectra task, the conceptually inaccurate, strong limitation of being just a prediction of a collection of energies and oscillator strengths with poor correlation, with no actual information and correlation with states’ character or wavefunction or orbitals involved, remains after revision.

We hope that this revision, introducing elucidation of excitation characters from

the OMNI-P2x predictions, addresses your concerns. We also note that massive simulations enabled by OMNI-P2x can serve for generating enough statistical data, from which representative samples can be drawn for additional manual analysis, or using post-processing tools such as Multwfn and TheoDORE mentioned below. Both our analysis tools, based on atom-wise contributions and wavefunction-analysis tools, are complementary. We added this discussion to the manuscript.

The means of analysis presented above do not exclude the need for dedicated analysis, either manual or using wavefunction-based tools such as TheoDORE[84] or Multiwfn[85], which can provide additional insights beyond spatial localization of the excitations, such as the orbitals' character. These tools are complementary, as massive-scale simulations enabled by OMNI-P2x can serve as a first step for generating enough statistical data, from where representative samples can be drawn, and post-processed further with the wavefunction-based tools.

The authors acknowledge this limitation and justify it in their response:

“While this limits interpretability in terms of electronic structure, it is consistent with how UV/Vis spectra are typically constructed from ab initio calculations, where oscillator strengths are reported for adiabatic states without explicit orbital labeling. The utility of OMNI-P2x is therefore in reproducing excitation spectra (energies and intensities) efficiently and at scale, rather than in providing mechanistic orbital analysis.”

We have now amended the manuscript with the newly introduced interpretation of the character of excited states with OMNI-P2x.

Here, I again disagree with the authors. Even if there is no explicit orbital labeling, an ab initio calculation includes molecular orbital and excited configuration coefficients that allow for determining the character of the excited states. Additionally, well-established post-processing codes like Multiwfn or TheoDORE can efficiently analyze wavefunctions and provide descriptors to train a machine learning model for excited states. This is a fundamental of computational spectroscopy, where the calculations are needed to explain what is behind the shape of the UV/Vis spectra. If this information is missing, the scope of computational chemistry is strongly diminished. That's why I believe that a ground-breaking architecture should be able to properly reproduce the spectrum because it reproduces correct energy, oscillator strengths, and wavefunction information, not just the shape. If this model deliberately ignores this aspect, it not only reduces the impact of computational chemistry but also its reliability in terms of the physical interpretation of the results.

We agree that post-processing tools like Multiwfn and TheoDORE are complementary to our approaches, particularly to the newly introduced analysis of the atom-wise contributions with OMNI-P2x. Analysis with OMNI-P2x of atom-wise contributions to excitations represents a significant step toward interpretable ML models for excited states, providing chemically meaningful insights about where excitations are localized. This addresses a key use case in computational spectroscopy: understanding which parts of a molecule are involved in electronic transitions, while maintaining the outstanding computational efficiency of OMNI-P2x. As mentioned above, both can be used together: Multiwfn and TheoDORE can be used for post-processing the single-point calculations done with the electronic structure methods on a fraction of structures generated from the OMNI-P2x simulations. An example would be taking representative snapshots from ML-propagated nonadiabatic dynamics for further analysis.

The authors write: “We note that, contrary to your comment, the $S_0 \rightarrow S_1$ excitation energies have consistently exhibited the lowest accuracy in both the original and revised models, so we are unsure where the claim of “the highest accuracy” originated.

I just wanted to clarify this point. If the character and order of the excited states are not the target of the prediction, then, in cases reported in Figure 5a,b,c, the prediction is quite acceptable. The shape and the position of the maxima are ok for a qualitatively low-cost prediction, so it is not a problem with the prediction of the lowest states.

We thank you for your clarification.

My comments about the NAMD example have been insufficiently addressed. I pointed out to the authors the reasons why this molecule was shown, implying a deeper physical analysis to justify the correctness of the simulation predicted. The trajectories are not analyzed to prove that the geometrical evolution of fulvene is well reproduced, passing through the correct conical intersections, and not only the time at which the hop should occur, or the energy gap. This is very important, especially because nonadiabatic couplings or wavefunction overlap are not predicted. I even suggested this point to the authors (since it is done in other implementations). I think this has not been properly addressed in the short paragraph in the discussion that was added.

We thank you for this suggestion; it is indeed one of the valuable ways to obtain additional analysis of the OMNI-P2x simulations. We have added the analysis of the geometric evolution of fulvene to the revised manuscript as described in our reply to the next comment.

Indeed, although the shape of the population resembles the CASSCF profile, there is no example of geometrical analysis, no check if the trajectory really passes through the correct conical intersection, or whether it's just the interpolation that correctly predicts the hops. The populations alone are not a metric of accuracy of a NAMD simulation, as highly sensitive to the simulation parameters, and they are not a measure of correct physical behaviour, as the decay from the excited states could be derived by a wrong deactivation channel. This is particularly important to show, especially because in the previous task on azobenzene, it was shown that out of the equilibrium, the prediction gets significantly worse. Same for the excited state character, there is no analysis to show that there is no problem in this example. Since in NAMD by definition geometries out of the ground state equilibrium are explored, and the excited states' character will change, this check is fundamental, in my opinion, to show the reliability of the model. It is missing the proof of the correct physical behaviour beyond just interpolation. Additionally, in figure 9, I believe that the y-axis should be the S_1 population..

We agree that state-averaged electronic state populations alone are not sufficient to judge the quality of ML-NAMD propagation.

We start by plotting the S_0 and S_1 PES, as simulated with the transfer learning ML model: panel (a) of Figure S13 of the ESI, and the CASSCF(6,6) reference, panel (b), along the two key reaction coordinates of fulvene C=CH₂ stretching and rotation:

The agreement between both the ML models, and the reference PES is excellent, with the fine-tuned model performing better further along the stretching coordinate.

As a next step, we analyze the minimum energy conical intersection (one of the many, at least 5, possible conical intersections of fulvene) optimized using both the ML model and the reference QC method, which we present in Figure S14 of the ESI.

We note an excellent agreement between both geometries, with an RMSD of 0.03 Å.

Finally, having confirmed that machine learning reproduces the correct PES topology, and MECI structure for fulvene, we confirm that the deactivation proceeds through physically correct deactivation channels. To do that, we analyze the hopping points in the reference CASSCF and ML-driven dynamics. A correlation plot between the C=CH₂ distance and the mean C=CH₂ dihedral angle at hopping geometries has been shown in Figure S15 of the ESI.

The distribution of the hopping geometries agrees excellently between the reference and ML-NAMD dynamics, showing the same pattern of early planar hopping, with later twisted-stretched or twisted-shrunk deactivation. The fractions of deactivation events proceeding through each channel have been quantified in Table S1, with ML-NAMD agreeing with the reference, within a 95% confidence interval, as shown in Table S2.

We also thank the reviewer for noticing the typo in the y-axis of Figure 9 (which should read S_0 population).

The results have been referenced in the manuscript as:

Further analysis of the NAMD trajectories of fulvene, including comparison of the ML-predicted potential energy surfaces, minimum energy conical intersections, as well as hopping points and deactivation channels, is provided in Section S4.1 of the ESI. In all cases, excellent agreement between the ML model and reference CASSCF calculations

is noted.

I appreciate the addition of a second experiment regarding the NAMD. However, the analysis is rather limited, and again, it is not shown, either in the SI, that the propagation that is delivered is correct. More importantly, also considering this example, I once again acknowledge the computational speed-up of the new active learning procedure for NAMD. But this is an improvement on how active learning is designed, but not a paradigm shift for how ML potentials are used in NAMD, since there is still a need for fine-tuning, and the model is not transferable for this task. Additionally, the lack of couplings, available in other architectures, does not push towards a physical improvement in the way MLIPs are used for NAMD.

We thank you for acknowledging our extension of the azobenzene transfer learning experiment. We agree that the need for fine-tuning is a computational burden that is not removed by the usage of OMNI-P2x, however, the construction of such a model is also fundamentally difficult from a computational chemistry perspective. Methods most commonly used for accurate and trustworthy NAMD propagation, such as MRCI, CASSCF, or perturbation theory methods like CASPT2, require selecting a system-specific active space, and hence cannot be directly replaced by a universal potential that has not been fine-tuned. For this reason, we believe that deriving a procedure that decreases the number of required points for stable and accurate NAMD propagation by a factor of 50 (in the case of azobenzene) is a groundbreaking methodological advancement.

Similar to the fulvene dynamics, we provide further analysis of the ML-predicted PES, as well as the MECI and deactivation pathways. Starting with the analysis of the PES, a scan around the C-N=N-C dihedral angle is presented in Figure S16 of the SI, which is the critical reaction coordinate for azobenzene photoisomerization. We note a very good agreement between the ML and reference PES, with an MAE of 0.04 eV for S0 and 0.03 eV for S1.

The MECI structures optimized with both the ML model and AIQM1 are presented in Figure S17. The structures agree well and represent the same physically correct “twisted-assisted inversion” mechanism.

Furthermore, an analysis of hopping geometries, presented as a correlation plot between the maximum CNN angle and the C-N=N-C dihedral angle (which are the two crucial reaction coordinates in this photoprocess), shows good agreement between ML-driven and reference NAMD trajectories, which is presented in Figure S18 of the ESI.

The manuscript has been adjusted to reference these results:

An analysis of azobenzene S0 and S1 PES', optimized MECIs, and deactivation geometries is provided in Section S4.2, along with a comparison to reference results, which agree excellently with the ML models.

In conclusion, while I agreed with the other reviewers about the advancement in terms of architecture, I also saw similar problems in the results shown. In my first revision, although I recognized that the model works, I underlined some important, potentially wrong interpretations that could be derived from the predictions of the model in computational chemistry applications. These have not been disproven, and the limitations have not been addressed, but remain after revision. Since I find them highly relevant, I still cannot recommend publication in Nature Communications, although I believe this version of the manuscript is improved.

We hope that this revision, based on your comments, has addressed your remaining reservations about this work.

Reviewer #3

The authors have given responses to each of my comments, and made substantial changes to the manuscript. Many of the concerns I and other referees had are addressed in a fully satisfactory manner, but some of the most important are not, as detailed below. Overall I would say the authors have demonstrated that the model shows some promise for applications in theoretical spectroscopy and non-adiabatic dynamics, and represents an interesting methodological step forward, and the ML model shows real promise.

We thank you for your evaluation that our work represents a methodological step forward and that the ML model shows real promise.

However, all reviewers have expressed a range of different concerns about its accuracy and the overselling of the capabilities of the current model. These concerns add up to an uncertainty that the model exhibits the level of accuracy that would be necessary for an out-of-the-box tool of genuine usefulness to practitioners not expert in analysis of the accuracy of ML models. Given that it has been presented as a ready-to-use general-purpose tool I am therefore worried that publication in Nature Comms would be giving a seal of approval on the accuracy of the model that the current results do not support.

We definitely agree that Nature Communications is a journal aiming to present tools and methods that push beyond the current state of the art, and at the same time, we believe this is exactly what our model is doing. We acknowledge that OMNI-P2x is far from perfect, but we clearly state the limitations in the manuscript, as well as provide uncertainty estimates that warn the user and give an idea about its applicability. Furthermore, multiple solutions to mitigate the disadvantages of OMNI-P2x, such as fine-tuning and active learning for NAMD, are presented in the manuscript. Hence, we believe the model can be useful to practitioners who are becoming increasingly skilled at utilizing ML tools and are aware of the accuracy and applicability of MLIPs. The work also

represents an important methodological advancement for specialists within the field of ML-assisted (photo)chemistry that will be relevant for the development of future models.

This is particularly important given the strong statements about the accuracy, some of which the authors have not significantly toned down despite referees concerns, and which it is very important to avoid being exaggerated. Examples include “making ML approaches for excited-states a viable alternative to established QM methods” and “The model enables out-of-the-box predictions of electronic spectra of organic molecules, with accuracy approaching TD-DFT/B3LYP”. Uncritical use of the model as a substitute for TD-DFT calculations would be extremely risky, given the current results. Therefore I would not support publication of the manuscript in its current form.

We have revised the statements about OMNI-P2x accuracy to better reflect the actual results, as well as possible applications of the model. The first statement has been revised to read:

...making ML approaches for excited-states a viable alternative to established, cost-efficient QM methods for high-throughput calculations, such as semi-empirical methods.

While the second has been amended to explicitly state the target and accuracy of the model:

The model enables out-of-the-box predictions of electronic spectra of organic molecules, targeting the TD-DFT/B3LYP level of theory, with an accuracy of about 0.15 eV. Importantly, OMNI-P2x offers higher-quality spectra predictions than commonly used semiempirical QM methods for excited states, while having a substantially lower computational cost.

Note that there is continued mixing of units even when 2 of the referees asked them not to do this as it is confusing for the reader. This unit mixing creates the appearance of deliberately obscuring the substantially worse performance of OMNI-P2x with respect to ground-state universal models, which if I am reading it correctly, appears to be a very substantially higher error (albeit with less training data). That is not consistent with statement that “they demonstrate that the ground-state error of OMNI-P2x exhibits decent performance, even with respect to the current state-of-the-art models”.

We apologize for the usage of multiple units. For other universal, ground-state models, we supply the error both in the originally reported unit, as well in kcal/mol. The exception was UMAs errors, which have been reported in eV/atom, which has a physically different meaning (quantifying the error in prediction per atom of molecule vs error of the total energy). The closest conversion available here is kcal/(mol*atom), which has been added to the manuscript. However, the same error metric was provided for OMNI-P2x ground-state energies as well.

We want to assure the reviewer that there was absolutely no intent to obscure our model's performance. Indeed, the RMSE of 2.94 kcal/mol (compared to, e.g., ANI-1, with an RMSE of 1.3 kcal/mol), and MAE of 1.6 kcal/mol (vs AIMNet2 1.25 kcal/mol for neutral, singlet systems) is higher, but we believe it is reasonable, given our training set uses equilibrium geometries from PubChemQC rather than the extensive conformational sampling in dedicated ground-state datasets. The performance gap primarily reflects these training data limitations, rather than any fundamental architectural constraints.

We have revised the imprecise statement mentioned in the comment to read:

This ground-state MAE can be compared to the similar metrics of other MLIPs: ANI-1 [68] with an RMSE of 1.3 kcal/mol (OMNI-P2x 2.93 kcal/mol), UMA [76] with around 1 to 3 meV/atom (0.023 to 0.069

kcal/(mol·atom), depending on the version (OMNI-P2x 0.11 kcal/(mol·atom)) and AIMNet2 [77] reporting a MAE of around 1.25 kcal/mol for neutral compounds. Although those results are not comparable one-to-one, as each of the models was trained and tested on a different dataset, they show that the ground-state error of OMNI-P2x remains within a reasonable range from current state-of-the-art models, despite not being the main focus of this study.

I find the decision to replace the model entirely with an ensemble with reduced errors at this stage somewhat surprising – has this substitution been made consistently across all the data in the paper, or just applied selectively to places where the referee concerns were most serious?

We thank you for your comment and provide further clarification. The practice of creating an ensemble of models is commonly used to increase the accuracy and stability of ML predictions, and has been applied in MLIPs such as AIQM1 or the ANI-1 family of models. Aside from the accuracy increase, this allows us to provide an uncertainty quantification, which, as mentioned further in this review, is a significant improvement and would not be possible with just one model. The new ensemble has been used consistently throughout the manuscript (and, as a consequence, e.g., Figure 5 shows different systems and spectra than in the original version). For fine-tuning experiments, one model (or two copies of the same model in the case of azobenzene screening) was used. Information about this has been added to the manuscript:

We have then fine-tuned two copies of one of the OMNI-P2x energy models on this set, using different training/validation splits...

and:

One model from the energy ensemble of OMNI-P2x was used for fine-tuning in all NAMD applications.

The statement “The excitation energy errors are similar in magnitude to the target state energy errors and follow the same trend” doesn’t really answer the original question of “should we expect errors on $S_0 \rightarrow S_x$ transition energies to exceed those of the contributing state?”

The excitation energies errors are expected to be slightly higher than the target state (S_x) energy (about 0.2-0.3 kcal/mol). A correlation plot between the excitation is presented below:

Figure 4a has been improved and is now more useful in determining the spread of errors.

We are glad that our revision was satisfactory.

Appropriate cautionary language has been inserted to address the problem of predicting as many as 10 excited states for small molecules, and the fact that increasing success may be due to increasing Rydberg character.

We are glad that our revision was satisfactory.

Minor presentational issues raised here have been acted on appropriately.

We are glad that our revision was satisfactory.

The main point of this comment was that the statement that the model approaches the accuracy of TDDFT is not justified by the data. The reason to use a ML surrogate model is to bypass the computational expense of a given high or moderate level of theory such as TDDFT or QC methods by replacing it with a model that gives an error compared its ground truth that is substantially lower than the error of the ground truth itself: otherwise one is simply compounding the ground truth error. The authors response is simply that the model makes a further error (on top of the TDDFT error) that is comparable to typical errors of TDDFT. I would say this is not acceptable in an ML surrogate model, and significantly undermines the possible uses of the model, and I would want to see the statement removed or revised substantially.

We agree with the reviewer that this statement might be potentially misleading and decided to remove it from the manuscript.

This section is more satisfactory now that the SPC spectra for the top 3 candidates is shown. These spectra show there is not particularly good agreement between OMNI-P2x spectra and TDDFT, but the presence of low-energy redshifted absorption in both does at least mean that the screening task of finding such molecules has been successfully achieved.

We are glad that our revision was satisfactory.

The new Section S4 of the ESI is very cryptic and does not add to the discussion. Taken at face value S12a simply compares LZBL with FSSH and shows they agree for the same PES, which is no great surprise and gives no information about the quality of the PES itself. Furthermore, Referee 2's concerns about the accuracy of the dynamics do not appear to have been fully addressed in the rebuttal. However, the demonstrations now included in Figure 10 are fairly convincing evidence that the TL based on OMNI-P2x is effective in significantly reducing the amount of training data required to converge the model predictions with respect to training set size in such calculations.

We thank you for the comments. Indeed, the reason for the inclusion of Section S4 into the ESI was to compare the effect of a coupling-free surface hopping scheme (LZBL), with FSSH that includes nonadiabatic couplings, as Reviewer 2 raised concerns about the accuracy of using a method that does not rely on nonadiabatic couplings: “This could be solved using another formulation of TSH, for which nonadiabatic coupling or wavefunction overlap would need to be predicted by the model...”. We agree that further analysis of the ML-TSH results is necessary, and provide it in section S4 of the manuscript, with a detailed discussion in response to Reviewer's #2 comment.

The fact that the revised model issues warnings when the predictions are highly uncertain (as assessed by committee standard deviation predictions, presumably) is a significant improvement on the original approach.

We thank you for acknowledging the positive impact of including an uncertainty quantification, which was made possible by employing ensemble averaging. Indeed, the committee standard deviation is used, which has been stated in the manuscript:

Additionally, OMNI-P2x provides an uncertainty quantification (UQ) of all predicted energies, using the standard deviations between quantities predicted by each of the three NN potentials as the error bar.

A warning is issued to the user if the UQ of any excited states exceeds a safe threshold, which is determined as the median plus three median absolute deviations of the UQs of the test set, for each electronic state.

Responses to the comments of the reviewers

Reviewer comments are in *italics*, and our responses are in **regular, blue font**.

Reviewer #2 (Remarks to the Author):

I thank the authors for addressing my comments and improving the manuscript, especially thanks to the new analysis reported.

In particular, the atom-wise energy decomposition and the geometrical analysis of fulvene dynamics make the interpretation of the results more reliable and robust. However, for the atom-wise energy composition analysis, a general metric to evaluate the accuracy would be more suitable than potentially cherry-picked examples. The comparison between B3LYP and ADC(2) is also important, showing that, although correlated, the performances of TD-DFT are systematically worse than ADC(2).

I think the manuscript is highly improved through the two revision rounds, and I have no additional comments for the authors that prevent me from recommending publication.

We thank you for your final positive evaluation and previous reviews.

Reviewer #3 (Remarks to the Author):

I do not have any new comments to add in relation to the latest revision.

We thank you for your previous reviews.

Reviewer #4:

We thank you for your contribution as a co-reviewer.